# The Implicit Bias of Gradient Descent toward Collaboration between Layers: A Dynamic Analysis of Multilayer Perceptions

**Zheng Wang**     **Geyong Min**
Department of Computer Science
University of Exeter
{zw360;G.Min}@exeter.ac.uk

**Wenjie Ruan**[*]
School of Computer Science
USTC
rwjie@ustc.edu.cn

## Abstract

The *implicit bias* of *gradient descent* has long been considered the primary mechanism explaining the superior *generalization* of over-parameterized neural networks without overfitting, even when the training error is zero. However, the implicit bias toward *adversarial robustness* has rarely been considered in the research community, although it is crucial for the *trustworthiness* of machine learning models. To fill this gap, in this paper, we explore whether layers in *neural networks* collaborate to strengthen adversarial robustness during gradient descent. By quantifying this collaboration between layers using our proposed concept, *co-correlation*, we demonstrate a monotonically increasing trend in co-correlation, which implies a decreasing trend in adversarial robustness during gradient descent. Additionally, we observe different behaviours between narrow and wide neural networks during gradient descent. We conducted extensive experiments that verified our proposed theorems.

## 1  Introduction

As Artificial Intelligence (AI) has been widely applied in many industrial sectors, understanding the theoretical properties behind modern machine learning models is important, especially for neural networks due to their black-box nature. One such property is *implicit bias*, stemming from the phenomenon where over-parameterized neural networks, trained in a *gradient descent* manner, often exhibit great generalization without over-fitting. This implicit bias of gradient descent is often explained as steering neural networks towards solutions characterized by max-margin [3, 23, 12].

Another intriguing phenomenon is the existence of adversarial examples — imperceptible perturbations of inputs that alter classification results. Apart from existing work on *attacks* — algorithms for generating adversarial examples [5, 39, 37], *defences* against attacks, e.g., adversarial training [25, 36] and distillation [28], and *verification* [40, 34] to identify safe regions guaranteeing the absence of adversarial examples, recently some works are aiming at a theoretical understanding behind adversarial robustness. However, a principled way to comprehend the core contributors to the vulnerability of neural networks, especially a theoretical understanding of their relation to generalization capabilities, remains fragmented. This fragmentation is largely due to the intricate nature of neural networks, where robustness is interconnected with many factors spanning across input data distribution [33, 14], sampling complexity [1, 27], optimization techniques [25], weight initialization strategies [41], and model capacity and architectures [32, 2, 18, 35]. Not to mention that only very few works can address both generalization and adversarial robustness in a uniform framework. One such research by Frei et al. [13] investigates the *implicit bias* concerning both *generalization* and *adversarial robustness*,

---

[*]Corresponding Author

asserting that while implicit bias leads to solutions with improved generalization, it results in weaker adversarial robustness. However, this work ignores the architectural factor of neural networks and is hard to generalize to neural networks with more than two layers.

Inspired by this work and to investigate whether neural network layers collaborate against adversarial examples during training, this paper first adopts the novel concept of Dirichlet energy—originating from Partial Differential Equations (PDEs) to assess the variability of a function [11]—to evaluate the adversarial robustness of neural networks. We then theoretically demonstrate that Dirichlet energy serves as an effective measure of adversarial risk. By decomposing the Dirichlet energy across the entire neural network into its constituent layers, we can quantify the interactions between adjacent layers concerning adversarial robustness. We term this interaction *collaboration correlation (co-correlation)* and find that this metric reflects *'alignments'* in feature selection between neighbouring layers. Furthermore, we conduct a dynamic analysis of co-correlation in two-layer MLPs, demonstrating with high probability a monotonic increase under gradient descent, which indicates diminishing adversarial robustness. Additionally, our experiments show that two-layer MLPs with small widths tend to enhance their performance through strengthened co-correlation, a pattern not observed in wide two-layer MLPs. Our key contributions in this paper can be summarized as follows:

1. To the best of our knowledge, this work is the first to study the implicit bias of interaction between layers. We have quantified the interactions between adjacent layers and theoretically demonstrated that co-correlation between layers strengthens during gradient descent in neural networks under mild assumptions, suggesting that it not only fails to collaborate against adversarial perturbations but may even hinder resistance to them during gradient descent.

2. We demonstrate how neural networks with a large width differ in behaviour from the neural network of a small width, showing that MLPs with larger widths exhibit more resistance to increased co-correlation and, therefore, are more adversarial robust, which is complementary for the work by Dohmatob and Bietti [8].

3. Extensive experiments have been conducted to validate our proposed framework. By controlling the weight initialization, a perspective also suggested by Zhu et al. [41], we challenge the argument, as proposed by Huang et al. [18], that a wide neural network does not necessarily lead to better adversarial robustness, through the lens of cross-layer collaboration.

## 2 Related Works

### 2.1 Implicit Bias of Gradient Descent

The mystery of over-parameterized neural network trained with *gradient descent* manner hardly over-fitting has long been studied. Chizat and Bach [3] study the two-layer neural network with infinite width and homogeneous activations, showing that gradient flow can be characterized as a max-margin classifier on exponentially tailed losses. Lyu et al. [24], Sarussi et al. [29] study the two-layer Leaky ReLU neural network on linearly separable data and claim that networks converge to a max-margin linear predictor by gradient descent manner. Frei et al. [12] confirms those claims on high-dimensional nearly orthogonal data.

Lyu and Li [23], Ji and Telgarsky [19] claims that the homogeneous neural networks with exponentially-tailed classification losses converge to a *KKT* point of a maximum-margin problem. Kunin et al. [21] extend these results to a more boarder family of *quasi-homogeneous* neural networks. A more recent research [13] considers both generalization and robustness for two-layer ReLU neural networks, arguing that gradient descent is biased towards solutions that generalise well but are more vulnerable against adversarial examples, even the neural network is highly over-parameterized.

### 2.2 Theoretical Investigation of Adversarial Robustness

Since the phenomenon of adversarial examples has been discovered [15], various works have been proposed to understand the theoretical fundamentals behind it, especially for neural networks. Some researchers argue that the source of adversarial vulnerability comes from the input data [33, 8, 31, 26, 14, 7, 30, 1, 27]. The more recent researches investigate the fragility of neural networks from an architectural perspective. Simon-Gabriel et al. [32] study the vulnerability of feed-forward neural

networks measured by $L_p$ norm of the loss function w.r.t. input data, suggesting that the vulnerability increases with input dimension independent of model structures. Daniely and Shacham [6] examined the ReLU neural network characterized by decreasing dimensions at each layer. They asserted that the manifestation of adversarial robustness is intrinsically tied to the network's architecture, which contrasts with the propositions put forth by Simon-Gabriel et al. [32]. Bubeck et al. [2] expanded the findings of Daniely and Shacham's work on two-layer neural networks from an "under-complete case" scenario to an "over-complete" one where the number of neurons surpasses the input dimension. They further broadened the conclusions drawn by Daniely and Shacham [6] and Bubeck et al. [2] to encompass Deep ReLU networks, hinting at a crucial role played by bottleneck layers in these networks. Zhu et al. [41], instead of merely considering random weights as the standard configuration, conducted a comprehensive analysis of the effects of weight initialization on adversarial robustness.

Unlike previous studies that focus solely on the overall assessment of neural networks while overlooking layer interactions, our research examines the synergistic involvement between layers within neural networks, taking into account both weight initialization and optimization.

## 3  Preliminary

### 3.1  General Setting

We follow the binary classification setting where the input data is $\mathcal{X} \subseteq \mathbb{R}^d$, with the label $\mathcal{Y} \subseteq \{0, 1\}$. Given data set $\mathcal{D} = \{(\boldsymbol{x}_i, y_i)\}_{i=1}^n$ drawn from an unknown probability measure $P$ on $\mathcal{X} \times \mathcal{Y}$ and the neural network $f : \mathcal{X} \times \Theta \to \mathbb{R}$, where $\Theta$ denotes the set of parameters, our objective is to optimize $f$ by updating the weights with *gradient descent* method such that it can predict the label accurately. The prediction result is shown in Equation (1).

$$y_{pred} = \begin{cases} 1, sig(f_W(\boldsymbol{x})) > 0.5 \\ 0, sig(f_W(\boldsymbol{x})) \le 0.5, \end{cases} \tag{1}$$

where $sig$ is the *sigmoid* function, i.e., $sig(x) = 1/(1 + e^{-x})$. We use *Binary Cross-Entropy (BCE) loss* in Equation (2) as our loss function. For simplicity, we denote $u_i = f(\boldsymbol{x}_i, W), i \in [n]$ as the output of the neural network for input $\boldsymbol{x}_i$, where $[n] = \{k \in \mathbb{N}^+ | k \le n\}$.

$$L(f, y) = \frac{1}{n} \sum_{i=1}^n L(sig(u_i), y_i) = -\frac{1}{n} \sum_{i=1}^n \left[ y_i \log(sig(u_i)) + (1 - y_i) \log(1 - sig(u_i)) \right]. \tag{2}$$

### 3.2  Neural Networks and Adversarial Risk

Our exploration starts from a basic linear model, then to *Multilayer Perceptrons (MLP)*. We provide the proof for both linear and 2-layer MLPs, which can be extended to MLPs with more layers. They are defined as

$$f_{linear}(\boldsymbol{x}, W) = \boldsymbol{a}^T(W\boldsymbol{x}) \tag{3a}$$

$$f_{mlp}(\boldsymbol{x}, W) = \boldsymbol{a}^T(\sigma(W\boldsymbol{x})), \tag{3b}$$

where $\boldsymbol{x} \in \mathcal{X}$ is the input data, $W \in \mathbb{R}^{m \times d}$ denotes the linear transformation, $m$ is the width of the networks. $\sigma$ denotes the element-wise *activation functions*.

We follow the initialization setting in [9], where $\boldsymbol{a}$ is randomly initialized and fixed from a binary selection of $\{-\frac{1}{\sqrt{m}}, \frac{1}{\sqrt{m}}\}^m$. Additionally, we introduce a slightly different setting for the weights $W$, which are randomly initialized following the normal distribution $N(0, \frac{1}{m^{1+2q}})$ with $q > 0$, instead of $N(0, \frac{1}{m})$. However, in our experiment, different settings of $q$ are considered, including the scenario that $q \le 0$.

Generalization ability is one of the most important concepts for machine learning models. Classifiers with better generalization power indicate lower *natural risk* for unseen data. Given data points $(\boldsymbol{x}, y) \sim P$ and classifier $f$, the natural risk is defined as

$$R(f) = \mathop{\mathbb{E}}_{(\boldsymbol{x}, y) \sim P} [L(f(\boldsymbol{x}), y))]. \tag{4}$$

When it comes to 0-1 loss, the natural risk becomes the probability of misclassification for unseen data points.

Similar to natural risk, the *adversarial risk* is defined as the probability of misclassification under adversarial perturbations as is shown in Definition 3.1.

**Definition 3.1** (Adversarial Risk). Given data points $(\boldsymbol{x}, y) \sim P$, and a perturbation $\boldsymbol{\varepsilon}$ within a norm-ball, i.e.,

$$B_r = \{\|\boldsymbol{\varepsilon}\|_2 \leq r\}, \tag{5}$$

where $r > 0$ indicates the $L_2$-norm budget for perturbations. The adversarial risk for neural network $f$ on loss function $L$ is defined as

$$R^{rob}(f, r) = \mathop{\mathbb{E}}_{(\boldsymbol{x}, y) \sim P} \left[ \sup_{\boldsymbol{\varepsilon} \in B_r} L(f(\boldsymbol{x} + \boldsymbol{\varepsilon}), y)) \right] \tag{6}$$

Since adversarial perturbations are almost invisible to human eyes, we expect $r$ to be quite small.

### 3.3 Dirichlet Energy

The concept of *Dirichlet energy*, originating from *Partial Differential Equations (PDEs)*, serves as a tool to assess the variability of a function [11]. However, as argued by Dohmatob and Bietti [8], it serves as a more effective measure of adversarial robustness than the *Lipschitz constant*. We extend this concept to mappings to make it more suitable for multi-dimensional problems, which is formally defined in Definition 3.2.

**Definition 3.2** (Dirichlet Energy for Mappings). Let convex set $\mathcal{Z}_1 \subseteq \mathbb{R}^{m_1}$ and $\mathcal{Z}_2 \subseteq \mathbb{R}^{m_2}$. Given a differentiable mapping $\boldsymbol{\phi} : \mathcal{Z}_1 \to \mathcal{Z}_2$, the Dirichlet Energy w.r.t. $\boldsymbol{x} \sim P_{\boldsymbol{x}}$ is defined as

$$\mathfrak{S}(\boldsymbol{\phi}) \triangleq \sqrt{\|J_{\boldsymbol{\phi}}(\boldsymbol{x})\|_{L^2(P_{\boldsymbol{x}})}^2} = \sqrt{\mathbb{E}_{\boldsymbol{x} \sim P_{\boldsymbol{x}}} [\|J_{\boldsymbol{\phi}}(\boldsymbol{x})\|_2^2]}, \tag{7}$$

where $\|\cdot\|_2$ refers to the operator norm and $J_{\boldsymbol{\phi}}(\boldsymbol{x})$ denotes the Jacobian matrix of $\boldsymbol{\phi}$ w.r.t. its input $\boldsymbol{x}$.

To be clear, we use $\|\cdot\|_2$ to denote the $L_2$-norm for vectors and operator norm for matrices throughout our analysis. This concept can be extended to layers with ReLU activation by extending the derivative to $ReLU'(x) = \mathbf{1}_{\{x \geq 0\}}$, where $\mathbf{1}_{\{x \geq 0\}}$ is the indicator function. With the concepts defined, we now establish the relationship between adversarial risk and Dirichlet energy.

## 4 Measure Adversarial Risk by Dirichlet Energy

In this section, we establish the relationship between the *adversarial risks* and *Dirichlet energy*, showing that Dirichlet Energy is a proper measurement for adversarial risk. Dohmatob and Bietti [8] only compares the Dirichlet energy with the Lipschitz constant. We, instead, illustrate that Dirichlet energy is a proper representation of the gap between natural risk and adversarial risk.

**Theorem 4.1.** *Given data points $(\boldsymbol{x}, y) \sim P$ and $\boldsymbol{x} \sim P_{\boldsymbol{x}}$, the relationship between adversarial risk and Dirichlet energy for classifier $f$ with differentiable loss function $L$ is shown as*

$$R^{rob}(f, r) \lesssim R(f) + r\mathfrak{S}(L(f)), \tag{8}$$

*where $r > 0$ is the largest perturbation budget and $\mathfrak{S}(L(f)) = \sqrt{\mathbb{E}_{\boldsymbol{x} \sim P_{\boldsymbol{x}}} \left[ \|\nabla_{\boldsymbol{f}} L^T \cdot J_{\boldsymbol{f}}(\boldsymbol{x})\|_2^2 \right]}$ indicating the Dirichlet energy of the classifier on loss $L$.*

As is shown, the Dirichlet energy is a proper representation of the gap between generalization and adversarial risk which indicates the adversarial robustness. The proof relies on the linear approximation of $L(f)$. The detailed proof is shown in Appendix A.

We also empirically show that the Dirichlet energy for classifier $f$, i.e., $\mathfrak{S}(f)$ instead of $\mathfrak{S}(L(f))$, is the part that influences the adversarial robustness, as is shown in Figure 1 where we compare the level of Dirichlet energy for $f$ with the *robust accuracy* of the classifier attacked by *Auto-attack* [5].

Since the Dirichlet energy defined in Equation (7) can be used to measure the *variability* of mappings, it follows that this metric could be employed to evaluate the adversarial robustness of individual layers or modules within neural networks. Consequently, this allows for an assessment of whether there is collaboration between these components in terms of adversarial robustness.

## 4.1 Measure Correlation between Layers

To evaluate the collective impact of neighbouring layers on the adversarial robustness, we regard the neural network as compounded separate mappings, i.e., $\phi \circ \varphi$. By calculating the Dirichlet energy for each mapping ($\mathfrak{S}(\phi)$ and $\mathfrak{S}(\varphi)$) and comparing it with the overall Dirichlet energy for the whole classifier ($\mathfrak{S}(\phi \circ \varphi)$), we can quantify the collaboration between these two mappings, and we term this quantification *co-correlation* which is defined in 4.2.

**Definition 4.2** (Co-correlation). Let $\varphi : \mathcal{Z}_1 \to \mathcal{Z}_2$ and $\phi : \mathcal{Z}_2 \to \mathcal{Z}_3$ be two successive mappings, where both $\mathcal{Z}_1$ and $\mathcal{Z}_2$ are convex. Given input $\boldsymbol{x} \sim P_{\boldsymbol{x}}$, the *collaboration correlation (co-correlation)* is defined as

$$\varrho_{\phi,\varphi} \triangleq \frac{\left(\mathbb{E}_{\boldsymbol{x} \sim P_{\boldsymbol{x}}} \|J_{\phi \circ \varphi}(\boldsymbol{x})\|_2^2\right)^{\frac{1}{2}}}{\left(\mathbb{E}_{\boldsymbol{x} \sim P_{\boldsymbol{x}}} \left[\|J_\phi(\varphi)\|_2^2 \cdot \|J_\varphi(\boldsymbol{x})\|_2^2\right]\right)^{\frac{1}{2}}}. \tag{9}$$

To avoid any confusion, we use $\varphi$ interchangeably to denote both the function $\varphi : \mathcal{Z}_1 \to \mathcal{Z}_2$ and the output of the function $\varphi = \varphi(\boldsymbol{x})$ for a given input $\boldsymbol{x}$. Consequently, $\varphi$ in $J_\phi(\varphi)$ represents the outputs, while in $J_\varphi(\boldsymbol{x})$, it represents the mapping. Since for each $\boldsymbol{x} \in P_{\boldsymbol{x}}$, we have

$$\|J_{\phi \circ \varphi}(\boldsymbol{x})\|_2 = \|J_\phi(\varphi) \cdot J_\varphi(\boldsymbol{x})\|_2 \leq \|J_\phi(\varphi)\|_2 \cdot \|J_\varphi(\boldsymbol{x})\|_2.$$

We expect $0 \leq \varrho_{\phi,\varphi} \leq 1$. The concept of co-correlation can be explained by the feature alignment of layers in neural networks.

**Interpretation of co-correlation.** Intuitively, co-correlation can be viewed as feature 'alignment' between layers. We illustrate this intuition with a 2-layer neural network with linear activation functions, i.e.,

$$f(\boldsymbol{x}) = \phi(\varphi(\boldsymbol{x})) = W_2 W_1 \boldsymbol{x}, \tag{10}$$

where $W_1$ and $W_2$ are weight matrices, and $\varphi(\boldsymbol{x}) = W_1 \boldsymbol{x}$ represents the feature selection in layer-1, while $\phi(\varphi) = W_2 \varphi$ represents it in layer-2. By Definition 4.2, the co-correlation between $\phi$ and $\varphi$ is

$$\varrho_{\phi,\varphi} = \frac{\|W_2 W_1\|_2}{\|W_2\|_2 \|W_1\|_2}. \tag{11}$$

where $\|\cdot\|_2$ denotes the operator norm. Let $\mathcal{S}_\varphi = \arg\max_{\|\boldsymbol{x}\|_2 = 1} \|W_1 \boldsymbol{x}\|_2$, representing the set of input features that maximize the $L_2$-norm of their corresponding outputs. Similarly, let $\mathcal{S}_\phi = \arg\max_{\|\boldsymbol{z}\|_2 = 1} \|W_2 \boldsymbol{z}\|_2$. Assume $\varrho_{\phi,\varphi} = 1$, there exists $\boldsymbol{x} \in \mathcal{S}_\varphi$ such that $\frac{\varphi(\boldsymbol{x})}{\|\varphi(\boldsymbol{x})\|_2} \in \mathcal{S}_\phi$. This implies that the maximal output in terms of the $L_2$-norm at the first layer can lead the output at the second layer to reach its maximum. In other words, in terms of maximizing outputs, layer 1 aligns with layer 2. We can thus regard co-correlation as the degree of alignment in feature selection between adjacent layers within the context of output maximization. In the non-linear case, the weight matrices are replaced by the Jacobians of the respective layers, which can be equivalently viewed as weight matrices that vary based on their inputs.

**Definition 4.3** (Other Related Statistics). Given the same assumptions in Definition 4.2, we define the linear correlation between $\|J_\phi(\varphi)\|_2$ and $\|J_\varphi(\boldsymbol{x})\|_2$ as

$$\rho_{\phi,\varphi} \triangleq \frac{\mathbb{E}_{\boldsymbol{x} \sim P_{\boldsymbol{x}}} \left[\|J_\phi(\varphi)\|_2 \cdot \|J_\varphi(\boldsymbol{x})\|_2\right]}{\left(\mathbb{E}_{\varphi \sim \varphi(\boldsymbol{x})} \|J_\phi(\varphi)\|_2^2\right)^{\frac{1}{2}} \left(\mathbb{E}_{\boldsymbol{x} \sim P_{\boldsymbol{x}}} \|J_\varphi(\boldsymbol{x})\|_2^2\right)^{\frac{1}{2}}}, \tag{12}$$

where $\varphi \sim \varphi(\boldsymbol{x})$ shows that $\varphi$ follows the distribution of $\varphi(\boldsymbol{x}), \boldsymbol{x} \sim P_{\boldsymbol{x}}$. It is obvious that $0 \leq \rho_{\phi,\varphi} \leq 1$ and when $\rho_{\phi,\varphi} = 1$, $\|J_\phi(\varphi)\|_2$ and $\|J_\varphi(\boldsymbol{x})\|_2$ are linear correlated.

The mean and variance for $\|J_\phi(\varphi)\|_2 \cdot \|J_\varphi(\boldsymbol{x})\|_2$ are defined as

$$\mu_{\phi,\varphi} \triangleq \mathbb{E}_{\boldsymbol{x} \sim P_{\boldsymbol{x}}} \left[\|J_\phi(\varphi)\|_2 \cdot \|J_\varphi(\boldsymbol{x})\|_2\right], \tag{13}$$

and

$$var_{\phi,\varphi} \triangleq Var_{\boldsymbol{x} \sim P_{\boldsymbol{x}}} \left[\|J_\phi(\varphi)\|_2 \cdot \|J_\varphi(\boldsymbol{x})\|_2\right]. \tag{14}$$

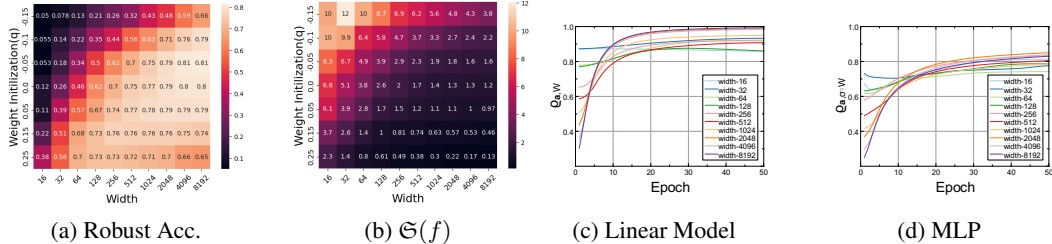

| (a) Robust Acc. | (b) $\mathfrak{S}(f)$ | (c) Linear Model | (d) MLP |

Figure 1: For all MLPs considered, lower value of Dirichlet Energy (Figure 1b) corresponds to larger robust accuracy on test-set attacked by $L_2$-norm Auto-attack with $\epsilon = 0.5$. The dynamics of the co-correlation $\varrho$ for linear and MLPs is shown in Figure 1c and 1d. The architectures of neural networks are defined in Equation (3b). The parameters to control the weight initialization in Assumption 5.1 is set to $q = 0.25$

*Remark* 4.4 (Linear Correlation for $L_2$-norm of the Jacobian). We have $\rho_{\phi,\varphi} = 1$ iff there exist $t \in \mathbb{R}$ such that

$$P(t\|J_\phi(\varphi)\|_2 = \|J_\varphi(\boldsymbol{x})\|_2) = 1,$$

implying that $\rho_{\phi,\varphi}$ certainly can be used to assess the linear correlations. It reduces to *Pearson correlation coefficient* [4] when the *mean* of both random variables equals zero.

Now we give the theorem that binds them together.

**Theorem 4.5** (Robustness Decomposition). *Given the same assumption in Definition 4.2, the measurement for overall adversarial robustness can be decomposed as*

$$\mathfrak{S}(\phi \circ \varphi) = \left(\mathbb{E}_{\boldsymbol{x} \sim P}\left[\|J_{\phi \circ \varphi}(\boldsymbol{x})\|_2^2\right]\right)^{\frac{1}{2}}$$

$$= \varrho_{\phi,\varphi}\left(1 + \frac{var_{\phi,\varphi}}{\mu_{\phi,\varphi}^2}\right)^{\frac{1}{2}} \rho_{\phi,\varphi}\mathfrak{S}(\phi)\mathfrak{S}(\varphi) \tag{15}$$

The proof is straightforward from the definition. Based on this theorem, we have conducted experiments with various linear models and 2-layer MLPs. These experiments demonstrate that, apart from the co-correlation, all other statistics are negligible, as shown in Figure 5 in the Appendix D. Consequently, our analysis primarily focuses on the co-correlation $\varrho$.

## 5 On Dynamics of Co-Correlation

### 5.1 Dynamics for Linear Model

Before delving into our analysis, we state our assumptions explicitly.

**Assumption 5.1.** We assume that each element $w_{i,j}$ in the weight matrix $W(0) \in \mathbb{R}^{m \times d}$ at initialization follows the Gaussian distribution $N(0, \frac{1}{m^{1+2q}})$, with $q > 0$. Additionally, each element $a_r, r \in [m]$ in $\boldsymbol{a}$ is randomly selected from the set $\{-\frac{1}{\sqrt{m}}, \frac{1}{\sqrt{m}}\}$, and fixed during training.

**Assumption 5.2.** We assume that for each $(\boldsymbol{x}_i, y_i) \in D, i \in [n]$, $\boldsymbol{x}_i$ is $L_2$ norm bounded such that $\|\boldsymbol{x}_i\|_2 = 1$ for all $i \in [n]$.

Since we only assume bounded inputs and a specific weight initialization method, compared to existing works [23, 19, 21, 12], our approach can be easily extended to MLPs with more than two layers.

Now let us focus on the co-correlation defined in Equation (9) and show the dynamics of $\varrho_{\phi,\varphi}$ for each step of gradient descent. We start from the linear model described in Equation (3a). Given the binary classification problem and the linear model described. Our first theorem demonstrates that co-correlation $\varrho_{\phi,\varphi}$ gradually accumulates throughout gradient descent optimization. Despite the simplicity of the linear model, it effectively exhibits most of the core properties under consideration.

Since the weights are updated by the gradient descent, the update of the weights at step $t \in \mathbb{N}$ is

$$\Delta \boldsymbol{w}_r(t) = \eta a_r \frac{1}{n} \sum_{i=1}^{n} \big(y_i - sig(u_i(t))\big) \boldsymbol{x}_i,$$

where $\eta$ denotes the learning rate. Therefore, the dynamics of the weights can be expressed as

$$\dot{W}(t) = \boldsymbol{a} \otimes \widetilde{\boldsymbol{x}}^T(t), \tag{16}$$

where $\otimes$ denotes the Kronecker product and $\widetilde{\boldsymbol{x}}(t)$ is the *error weighted input* such that

$$\widetilde{\boldsymbol{x}}(t) = \frac{1}{n} \sum_{i=1}^{n} \Big[ y_i - sig\big(u_i(t)\big) \Big] \boldsymbol{x}_i. \tag{17}$$

Given the weight updates, we demonstrate that the dynamics of the co-correlation for the linear model, denoted as $\dot{\varrho}_{\boldsymbol{a},W}(t)$, exhibit an increasing trend, particularly during the initial steps of training when most predictions are still essentially random.

**Theorem 5.3** (Dynamics of the Co-correlation for Linear Model). *Given the linear model defined in Equation* (3a) *and training dataset* $\mathcal{D} = \{(\boldsymbol{x}_i, y_i)\}_{i=1}^{n}$. *Assume that assumptions 5.1 and 5.2 hold for $W$ and $\boldsymbol{a}$. The gradient descent applied to the weights results in the dynamics of the co-correlation being expressed as:*

$$\dot{\varrho}_{\boldsymbol{a},W}(t) = \eta C(t) \varrho_{\boldsymbol{a},W}, \tag{18}$$

*and with high probability,*

$$C(t) \geq \frac{\sum_{\tau=1}^{t} \widetilde{\boldsymbol{x}}(\tau)^T \widetilde{\boldsymbol{x}}(t)}{\|W(t)\|_2^2} \cdot \Big(1 - \big(\boldsymbol{v}(t)^T \boldsymbol{a}\big)^2\Big) + \mathcal{O}\Big(\frac{1}{m^q}\Big) \tag{19}$$

*where the $\boldsymbol{v}(t)$ is the dominate eigenvector for $W(t)W(t)^T$.*

*When $m$ is sufficiently large, and during the initial steps of the optimization process, $\widetilde{\boldsymbol{x}}(\tau), \tau \in [t]$ are quite similar to each other in terms of cosine similarity, implying an acute angle to each other, which leads to $\sum_{\tau=1}^{t} \widetilde{\boldsymbol{x}}(\tau)^T \widetilde{\boldsymbol{x}}(t) \geq 0$. As a result, we can conclude that $C(t) \geq 0$.*

The detailed proof can be found in the Appendix B. This assertion is also corroborated by the results of our experiments as shown in Figure 1c. Even though Theorem 5.3 is based on a linear model, the essential properties are universally applicable and can be summarized as follows:

**Property 1.** The co-correlation $\varrho_{\boldsymbol{a},W}$ develops during the initial stages of training and becomes saturated as training progresses to its later stages.

**Property 2.** The speed of the accumulation of co-correlation $\varrho_{\boldsymbol{a},W}$ is inversely related to the operator norm of weights $\|W(t)\|_2$.

Under the same weight initialization conditions specified in Assumption 5.1, an increase in network width leads to a decrease in the $L_2$-norm of the weight, consequently causing a substantial rise in co-correlation.

## 5.2 Dynamics for MLP Model

For the non-linear case, we make certain assumptions regarding activation functions.

**Assumption 5.4.** The derivative of the activation function $\sigma'(x)$ in non-linear neural networks is bounded by $M$. In other words, we have $|\sigma'(x)| \leq M$.

With Assumption 5.4, Theorem 5.3 can be extended to MLPs, and the two properties still hold. Different from the linear model, the update of weights for non-linear MLP defined in (3b) is

$$\Delta \boldsymbol{w}_r = \eta a_r \frac{1}{n} \sum_{i=1}^{n} \big(y_i - sig(u_i)\big) \sigma'(\boldsymbol{w}_r^T \boldsymbol{x_i}) \boldsymbol{x}_i, \tag{20}$$

where $\sigma'(\boldsymbol{w}_r^T \boldsymbol{x_i})$ is the derivative of activation function w.r.t. its input. Hence,

$$\Delta W = \eta \begin{pmatrix} a_1 \widetilde{\boldsymbol{x}}_1^T \\ \vdots \\ a_m \widetilde{\boldsymbol{x}}_m^T \end{pmatrix},$$

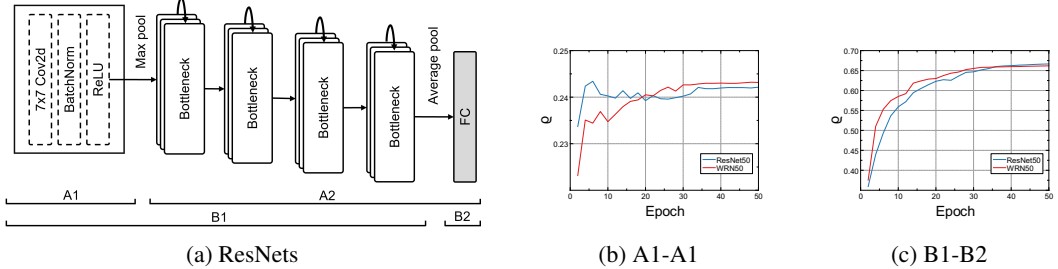

|   (a) ResNets   |   (b) A1-A1   |   (c) B1-B2   |

Figure 2: The dynamics of co-correlation for ResNet50 and WRN50 under different way of partition. The way of partition is illustrated in Figure 2a. A1-A2 and B1-B2 represent the separations that distinguish the head and tail separately.

where for each $r \in [m]$,

$$\widetilde{\boldsymbol{x}}_r = \frac{1}{n} \sum_{i=1}^{n} \sigma'(\boldsymbol{w}_r^T \boldsymbol{x_i})\big(y_i - sig(u_i)\big)\boldsymbol{x}_i.$$

As is shown in Equation (20), $\boldsymbol{a}$ is not the dominant eigenvector for $\Delta W \Delta W^T$ due to the difference of $\widetilde{\boldsymbol{x}}_r, r \in [m]$ from $\widetilde{\boldsymbol{x}}$. Thus to average out the difference, we define the weighted sum of inputs for the non-linear model as

$$\widetilde{\boldsymbol{x}}_*(t) \triangleq \frac{1}{n} \sum_{i=1}^{n} \alpha_i(t, \boldsymbol{x})\big(y_i - sig(u_i(t))\big)\boldsymbol{x}_i, \tag{21}$$

where $(\boldsymbol{x}_i, y_i) \in \mathcal{D}$ are realized r.v., and $\boldsymbol{x} \sim P_{\boldsymbol{x}}$. Given $w_{i,j} \overset{i.i.d.}{\sim} N(0, \frac{1}{m^{1+2q}})$ for each element in $W(0)$, we have $\boldsymbol{w}(0) \sim N(0, \frac{1}{m^{1+2q}}\mathbf{I}_d)$. Furthermore, given that $W(t)$ is calculated from $W(0)$, it is evident that each row in $W(t)$, denoted as $\boldsymbol{w}(t)$, constitutes a random variable. As a result, both $\sigma'(\boldsymbol{w}(t)^T \boldsymbol{x})$ and $\sigma'(\boldsymbol{w}(t)^T \boldsymbol{x}_i)$ are bounded random variables that are contingent upon $W(0)$. Hence, we define $\alpha_i(t, \boldsymbol{x})$ as

$$\alpha_i(t, \boldsymbol{x}) \triangleq \mathbb{E}_{W(0)}\Big[\sigma'(\boldsymbol{w}(t)^T \boldsymbol{x})\sigma'(\boldsymbol{w}(t)^T \boldsymbol{x}_i)\Big].$$

Since, $\boldsymbol{x} \sim P_{\boldsymbol{x}}$, $\alpha_i(t, \boldsymbol{x})$ is still r.v. contingent to $\boldsymbol{x}$ and so does the $\widetilde{\boldsymbol{x}}_*(t)$. Now we show the dynamics of co-correlation for two-layer MLP defined in Equation (3b).

**Theorem 5.5.** *(Dynamics of the Co-correlation for MLP) Given the MLP defined in Equation (3) with training dataset $\mathcal{D} = \{(\boldsymbol{x}_i, y_i)\}_{i=1}^{n}$, $\boldsymbol{x} \in \mathcal{X}$ such that $\boldsymbol{x} \sim P_{\boldsymbol{x}}$. Assume that Assumption 5.1 and 5.2 hold for $W$ and $\boldsymbol{a}$, and Assumption 5.4 holds for the activation function. we have*

$$\dot{\varrho}_{\boldsymbol{a}, \sigma \circ W}(t) = \eta C(t)\varrho_{\boldsymbol{a}, \sigma \circ W}(t).$$

*With high probability,*

$$C(t) \geq \frac{\sum_{\tau=1}^{t}\big(1 - \boldsymbol{a}^T \boldsymbol{v}(\tau)\boldsymbol{a}^T \boldsymbol{v}(t)\big)\mathbb{E}_{\boldsymbol{x} \sim P_{\boldsymbol{x}}}\big[\widetilde{\boldsymbol{x}}_*^T(\tau)\widetilde{\boldsymbol{x}}_*(t)\big]}{\mathbb{E}_{\boldsymbol{x} \sim P_{\boldsymbol{x}}}\|D(t)W(t)\|_2^2} + \max\left\{\mathcal{O}\left(\frac{1}{\sqrt{m}}\right), \mathcal{O}\left(\frac{1}{m^q}\right)\right\},$$

*where*

$$D(t) = diag(\sigma'(\boldsymbol{w}_1(t)^T \boldsymbol{x}), \cdots, \sigma'(\boldsymbol{w}_m(t)^T \boldsymbol{x})),$$

*and $\boldsymbol{v}(t)$ denotes the dominant eigenvector for $W(t)W(t)^T$, with $\widetilde{\boldsymbol{x}}_*^T$ is defined in Equation (21). Similar to the Theorem 5.3, when $m$ is sufficiently large, and during the initial steps of the optimization where the error-weighted inputs $\widetilde{\boldsymbol{x}}_*^T(\tau), \tau \in [t]$ do not significantly fluctuate, we have that $C(t) \geq 0$.*

The detailed proof is in Appendix C. In Theorem 5.5, $\widetilde{\boldsymbol{x}}_*$ serves a similar purpose as $\widetilde{\boldsymbol{x}}$ for the linear model. In addition, it considers the influence of the activation function. Property 1 still holds for the MLPs, and Property 2 extends to $\mathbb{E}_{\boldsymbol{x} \sim P_{\boldsymbol{x}}}\|D(t)W(t)\|_2^2 = \|J_{\sigma \circ W}(\boldsymbol{x})\|_{L(P_{\boldsymbol{z}})}^2$.

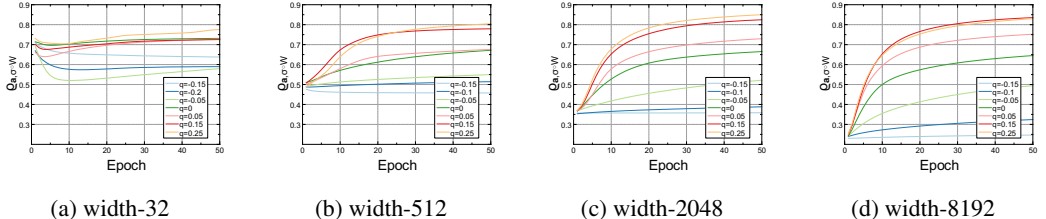

| (a) width-32 | (b) width-512 | (c) width-2048 | (d) width-8192 |

Figure 3: The dynamic of co-correlation under different set-up of weight initialization. MLP network defined in Equation (3) with ReLU activation function for width 32, 512, 2048 and 8192 are included.

## 6   Experiments

To estimate the co-correlation more efficiently and in parallel, we employ the *Power Iteration* algorithm [10] in conjunction with *Functorch* [17]. The corresponding pseudo-code is presented in the Appendix E.

We verify our proposed theorem on linear and MLP models on the MNIST dataset [22]. The width of hidden layers varied from $2^4$ to $2^{13}$, with weights initialized via a Gaussian distribution $N(0, \frac{1}{m^{1+2q}})$ where $q$ was set to values ranging from $0.25$ to $-0.15$. We train both the linear and MLP for $50$ epochs with a batch size of $512$ using the SGD optimizer with a learning rate of $0.003$. In addition, we also conduct the experiment on more complex ResNet50 [16] and WRN50 [38]. For both models, we opted for default random weight initialization and used the Adam optimizer with a learning rate of $0.0005$ on CIFAR10 [20]. To calculate the co-correlation and other statistics, we cover the entire testset. We consider using $L_2$ Auto-attack [5] with $\epsilon = 0.5$ for all MLPs we trained. The experiments were executed on a Nvidia RTX3090 GPU, using Python 3.9.7 and PyTorch 1.9.1. The code for the experiment is available at `https://github.com/squarewang2077/co-correlation`.

### 6.1   Empirical Evidence for Proposed Theorem

Figure 1 presents a comparison between the robust accuracy and the Dirichlet energy $\mathfrak{S}(f)$ across all trained MLPs. As observed in Figure 1a and 1b, models with lower levels of Dirichlet energy $\mathfrak{S}(f)$ tend to exhibit higher robust accuracy, suggesting that Dirichlet energy is an effective representation of adversarial robustness. Another noteworthy finding is that wider neural networks, with the same level of weight initialization, demonstrate improved adversarial robustness.

Figure 1 also depicts the dynamic behaviour of shallow neural networks with a weight initialization parameter of $q = 0.25$. As is shown in Figure 1c and 1d, the co-correlation $\varrho$ increases throughout training. Except for narrow widths like $2^4$ and $2^5$, the majority of networks demonstrate an upward trend. This trend, however, flattens for non-linear models, suggesting potentially stronger adversarial robustness due to the non-linearity of the activation function introduced in MLPs.

Figure 2 shows the dynamics of the co-correlation on ResNet50 and Wide-ResNet50. Both networks are trained on CIFAR10 using the Adam optimizer. We divide them by the pattern of A1-A2 and B1-B2, as shown in Figure 2a. The co-correlation outcomes for these divisions are displayed in Figure 2b and Figure 2c, it shows that even with the Adam optimizer, without specific weight initialization considerations, there is a noticeable rise in co-correlation.

### 6.2   The Impact of Width and Weight Initialization

Figure 3 illustrates the co-correlation dynamics under varying $q$ for both linear and MLPs with widths of 32, 512, 2,048, and 8,192. The figure highlights that our proposed theorems' assumption of $q > 0$ is quite tight, as all trajectories with $q < 0$ remain flat throughout training. We can also observe that the speed of accumulation significantly increases with larger network widths.

Figure 4 displays the accuracy on testset and co-correlation for both linear and MLPs as heat-maps. Each cell in the heat-map represents a trained network. From Figure 4c and Figure 4d, we observe that the best performance and robustness are shown by the MLPs with the largest widths (width $= 8192$) and the smallest weight initializations ($q = -0.15$). And when we alter the weight initialization to

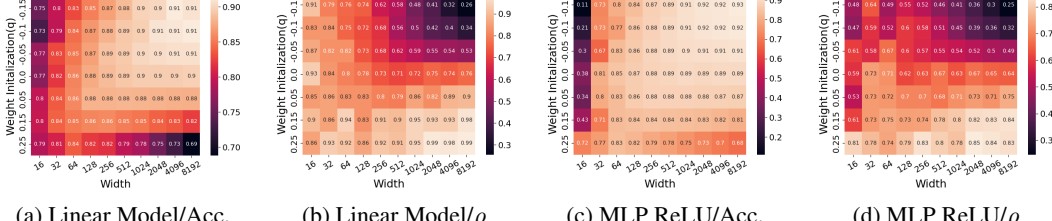

|                    |                    |                    |                    |
| :----------------: | :----------------: | :----------------: | :----------------: |
| (a) Linear Model/Acc. | (b) Linear Model/$\varrho$ | (c) MLP ReLU/Acc. | (d) MLP ReLU/$\varrho$ |

Figure 4: Accuracy and Co-correlation under different initialization and width. Figure 4a and 4b show the heat map for linear model, and Figure 4c and 4d is for MLP ReLU.

control the co-correlation, making it increase from $0.25$ to $0.83$, the accuracy declines accordingly from $0.93$ to $0.68$. On the contrary, for another extreme case of models with a width of $16$, enhanced performance is accompanied by increased co-correlation. Consequently, an interesting conclusion can be drawn about the diverse behaviour of neural networks with small and larger width. Gradient descent tends to enhance the training of neural networks with smaller width by fostering co-correlation among layers, which is intrinsically brittle. However, wide networks are trained with less reliance on interlayer correlation, resulting in inherently more robust models.

## 7    Conclusion and Limitation

Our work investigates the implicit bias of gradient descent toward adversarial robustness from the perspective of collaboration between layers. By adapting Dirichlet energy to estimate the adversarial robustness of neural networks' individual components, we characterized the collaboration behaviour between consecutive layers and identified two fundamental properties for dynamics of the co-correlation. The first property shows that the co-correlation for MLPs will build up during gradient descent under mild assumptions for weight initialization. The second property shows that the speed of accumulation for co-correlation is inversely related to the operator norm of Jacobian for the corresponding sub-modules. In addition, we observed that networks with small widths tend to foster co-correlation among layers to improve performance, whereas wide networks' performance improvement does not heavily rely on establishing such co-correlation. Future research can expand upon this by examining the effects of increased network depth and more sophisticated structures on the observed phenomena.

**Limitation**    Our work can be easily extended to multi-layer neural networks since we only assume that the inputs are bounded by the $L_2$-norm. However, like many theoretical studies, extending our approach to more complex models is challenging. It remains unknown whether complex models exhibit the same behaviors.

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

# A The Proof for Theorem 4.1

Suppose that the loss function $L$ is differentiable w.r.t. $f$. Given data points $(\boldsymbol{x}, y) \sim P$ and $\boldsymbol{x} \sim P_{\boldsymbol{x}}$, we have

$$R^{rob}(f, r) = \mathop{\mathbb{E}}_{(\boldsymbol{x},y)\sim P} \sup_{\varepsilon \in B_r} L(f(\boldsymbol{x} + \boldsymbol{\varepsilon}), y)) \tag{22}$$

$$= \mathop{\mathbb{E}}_{(\boldsymbol{x},y)\sim P} \left[ \sup_{\varepsilon \in B_r} L(f(\boldsymbol{x}), y)) + \sup_{\varepsilon \in B_r} L(f(\boldsymbol{x} + \boldsymbol{\varepsilon}), y)) - \sup_{\varepsilon \in B_r} L(f(\boldsymbol{x}), y)) \right] \tag{23}$$

$$\leq \mathop{\mathbb{E}}_{(\boldsymbol{x},y)\sim P} [L(f(\boldsymbol{x}), y))] + \sqrt{\mathop{\mathbb{E}}_{(\boldsymbol{x},y)\sim P} \left[ \sup_{\varepsilon \in B_r} \left(L(f(\boldsymbol{x} + \boldsymbol{\varepsilon}), y)) - L(f(\boldsymbol{x}, y))\right)^2 \right]} \tag{24}$$

$$\approx \mathop{\mathbb{E}}_{(\boldsymbol{x},y)\sim P} [L(f(\boldsymbol{x}), y))] + \sqrt{\mathop{\mathbb{E}}_{(\boldsymbol{x},y)\sim P} \left[ \sup_{\varepsilon \in B_r} \left(\nabla_f L^T \cdot J_f(\boldsymbol{x})\boldsymbol{\varepsilon}\right)^2 \right]} \tag{25}$$

$$= \mathop{\mathbb{E}}_{(\boldsymbol{x},y)\sim P} [L(f(\boldsymbol{x}), y))] + r\sqrt{\mathop{\mathbb{E}}_{(\boldsymbol{x},y)\sim P} \|\nabla_f L^T \cdot J_f(\boldsymbol{x})\|_2^2}, \tag{26}$$

where $\nabla_f L$ is the gradients of $L$ w.r.t. $f$, $\nabla_f L^T$ indicates that it is a row vector. $J_f(\boldsymbol{x})$ is the Jacobian matrix for classifier $f$.

# B The Proof of Theorem 5.3

Before formally proving the theorem, we provide some lemmas which are useful for our proof.

**Lemma B.1** (Dynamic of Weights for Linear Model). *Given gradient descent to optimize the weights, the dynamic of the weights at step $t$ is*

$$\dot{W}(t) = \boldsymbol{a} \otimes \tilde{\boldsymbol{x}}^T(t), \tag{27}$$

*Proof.* Given the step size $\eta$, the update for $r_{th}$ row of the weight matrix $W$ is

$$\Delta \boldsymbol{w}_r = -\frac{\eta}{n} \sum_{i=1}^{n} \frac{\partial l}{\partial u_i} \frac{\partial f(W, \boldsymbol{x}_i)}{\partial \boldsymbol{w}_r} \tag{28}$$

$$= \eta a_r \frac{1}{n} \sum_{i=1}^{n} \left(y_i - sig(u_i)\right) \boldsymbol{x}_i, \tag{29}$$

where $sig(\cdot)$ denotes the $sigmoid$ function. Hence, for weight matrix $W$, we have

$$\Delta W = \begin{pmatrix} \Delta \boldsymbol{w}_1^T \\ \vdots \\ \Delta \boldsymbol{w}_m^T \end{pmatrix} \tag{30}$$

$$= \eta \begin{pmatrix} a_1 \frac{1}{n} \sum_{i=1}^{n} \left(y_i - sig(u_i)\right) \boldsymbol{x}_i^T \\ \vdots \\ a_m \frac{1}{n} \sum_{i=1}^{n} \left(y_i - sig(u_i)\right) \boldsymbol{x}_i^T \end{pmatrix}. \tag{31}$$

After replacing with

$$\tilde{\boldsymbol{x}} = \frac{1}{n} \sum_{i=1}^{n} \left(y_i - sig(u_i)\right) \boldsymbol{x}_i, \tag{32}$$

we have

$$\Delta W = \eta \begin{pmatrix} a_1 \widetilde{\boldsymbol{x}}^T \\ \vdots \\ a_m \widetilde{\boldsymbol{x}}^T \end{pmatrix} \tag{33}$$

$$= \eta \boldsymbol{a} \otimes \widetilde{\boldsymbol{x}}^T. \tag{34}$$

Since $\|\boldsymbol{x}_i\|_2 = 1$, we also have $\|\widetilde{\boldsymbol{x}}\|_2 \leq 1$. Therefore, we can say that the dynamics of weights, i.e. $\dot{W}(t)$, is

$$\dot{W}(t) = \boldsymbol{a} \otimes \widetilde{\boldsymbol{x}}^T(t). \tag{35}$$

$\square$

**Lemma B.2** (The Concentration for $L_2$ Norm of Gaussian r.v.). *Let $I_m$ be identical matrix of size $m \times m$. Given $n$ Gaussian random vectors $\boldsymbol{z}_1, \cdots, \boldsymbol{z}_n$ such that $\boldsymbol{z}_i \overset{i.i.d.}{\sim} N(0, \frac{1}{m^{1+2q}}\mathbf{I}_m), q > 0, \forall i \in [n]$. With probability at least $1 - \delta$, we have several conclusions:*

*1. Average of the norm*

$$\frac{1}{n} \sum_{i=1}^{n} \|\boldsymbol{z}_i\|_2 = \frac{1}{m^q} + \mathcal{O}\left(\sqrt{\frac{8\log(2n/\delta)}{m^{1+2q}}}\right). \tag{36}$$

*2. Average of the square norm*

$$\frac{1}{n} \sum_{i=1}^{n} \|\boldsymbol{z}_i\|_2^2 = \frac{1}{m^{2q}} + \mathcal{O}\left(\max\left\{\frac{8\log(2/\delta)}{m^{1+2q}n}, \sqrt{\frac{8\log(2/\delta)}{m^{1+4q}n}}\right\}\right). \tag{37}$$

*3. Square root of the average of the square norm*

$$\sqrt{\frac{1}{n} \sum_{i=1}^{n} \|\boldsymbol{z}_i\|_2^2} = \frac{1}{m^q} + \mathcal{O}\left(\sqrt{\frac{8\log(2/\delta)}{m^{1+2q}n}}\right). \tag{38}$$

*Proof.* We first prove the concentration property for the average of the norm. Since $\sqrt{m^{1+2q}}z_{i,r} \overset{i.i.d.}{\sim} N(0,1), i \in [n], r \in [m]$, we have that $m^{1+2q}z_{i,r}^2 \sim \chi^2(1)$. Denote the sub-Exponential distribution as $SE(\nu^2, \alpha)$, we have $m^{1+2q}z_{i,r}^2 - 1 \in SE(4, 4)$. Hence $\forall i \in [n]$,

$$\mathbb{P}\left(\left|\frac{1}{m} \sum_{r=1}^{m} \left(\sqrt{m^{1+2q}}z_{i,r}\right)^2 - 1\right| \geq \epsilon\right) \leq \begin{cases} 2\exp\left(-\frac{m\epsilon^2}{8}\right) & \epsilon \in (0,1) \\ 2\exp\left(-\frac{m\epsilon}{8}\right) & \epsilon \geq 1 \end{cases}. \tag{39}$$

Therefore, we have $\forall \epsilon > 0$

$$\mathbb{P}\left(\left|\frac{1}{\sqrt{m}}\|\sqrt{m^{1+2q}}\boldsymbol{z}_i\|_2 - 1\right| \geq \epsilon\right) \leq \mathbb{P}\left(\left|\frac{1}{m}\|\sqrt{m^{1+2q}}\boldsymbol{z}_i\|_2^2 - 1\right| \geq \max\{\epsilon, \epsilon^2\}\right) \tag{40}$$

$$\leq 2\exp\left(-\frac{m\epsilon^2}{8}\right). \tag{41}$$

Equation (40) is because of the fact that given $c > 0$ fixed, $\forall x > 0$ we have

$$|x - 1| \geq c \Rightarrow |x^2 - 1| \geq \max\{c, c^2\} \tag{42}$$

Hence,

$$\mathbb{P}\left(\left|\frac{1}{n}\sum_{i=1}^{n}\|m^q z_i\|_2 - 1\right| \geq \epsilon\right) = \mathbb{P}\left(\left|\frac{1}{n}\sum_{i=1}^{n}\frac{1}{\sqrt{m}}\|\sqrt{m^{1+2q}}z_i\|_2 - 1\right| \geq \epsilon\right) \tag{43}$$

$$\leq \mathbb{P}\left(\sum_{i=1}^{n}\left|\frac{1}{\sqrt{m}}\|\sqrt{m^{1+2q}}z_i\|_2 - 1\right| \geq n\epsilon\right) \tag{44}$$

$$\leq \sum_{i=1}^{n}\mathbb{P}\left(\left|\frac{1}{\sqrt{m}}\|\sqrt{m^{1+2q}}z_i\|_2 - 1\right| \geq \epsilon\right) \tag{45}$$

$$\leq 2n\exp\left(-\frac{m\epsilon^2}{8}\right) \tag{46}$$

Equivalently, with probability at least $1 - \delta$,

$$\left|\frac{1}{n}\sum_{i=1}^{n}m^q\|z_i\|_2 - 1\right| \leq \sqrt{\frac{8}{m}log\frac{2n}{\delta}} \tag{47}$$

$$\Rightarrow \left|\frac{1}{n}\sum_{i=1}^{n}\|z_i\|_2 - \frac{1}{m^q}\right| \leq \frac{1}{m^q}\sqrt{\frac{8}{m}log\frac{2n}{\delta}} \tag{48}$$

To prove the second convergence, we starts from the fact that

$$\frac{1}{n}\sum_{i=1}^{n}\|z_i\|_2^2 = \frac{1}{n}\sum_{i=1}^{n}\sum_{r=1}^{m}z_{i,r}^2 = \frac{1}{m^{2q}}\frac{1}{nm}\sum_{i=1}^{n}\sum_{r=1}^{m}\left(\sqrt{m^{1+2q}}z_{i,r}\right)^2. \tag{49}$$

Since $\left(\sqrt{m^{1+2q}}z_{i,r}\right)^2 \overset{i.i.d.}{\sim} \chi^2(1), \forall i \in [n], r \in [m]$, similar to Equation (39), we have

$$\mathbb{P}\left(\left|\frac{1}{nm}\sum_{i=1}^{n}\sum_{r=1}^{m}\left(\sqrt{m^{1+2q}}z_{i,r}\right)^2 - 1\right| \geq \epsilon\right) \leq \begin{cases} 2\exp\left(-\frac{nm}{8}\epsilon^2\right) & \epsilon \in (0,1) \\ 2\exp\left(-\frac{nm}{8}\epsilon\right) & \epsilon \geq 1 \end{cases} \tag{50}$$

Equivalently, with probability at least $1 - \delta$,

$$\left|\frac{1}{n}\sum_{i=1}^{n}m^{2q}\|z_i\|_2^2 - 1\right| \leq \begin{cases} \sqrt{\frac{8}{mn}log\frac{2}{\delta}} & \epsilon \in (0,1) \\ \frac{8}{mn}log\frac{2}{\delta} & \epsilon \geq 1 \end{cases} \tag{51}$$

$$\Rightarrow \left|\frac{1}{n}\sum_{i=1}^{n}\|z_i\|_2^2 - \frac{1}{m^{2q}}\right| \leq \frac{1}{m^{2q}}\max\left\{\frac{8}{mn}log\frac{2}{\delta}, \sqrt{\frac{8}{mn}log\frac{2}{\delta}}\right\} \tag{52}$$

Now we prove the third concentration, with similar trick of the first concentration, we have

$$\mathbb{P}\left(\left|\sqrt{\frac{1}{nm}\sum_{i=1}^{n}\sum_{r=1}^{m}\left(\sqrt{m^{1+2q}}z_{i,r}\right)^2} - 1\right| \geq \epsilon\right) \leq \mathbb{P}\left(\left|\frac{1}{nm}\sum_{i=1}^{n}\sum_{r=1}^{m}\left(\sqrt{m^{1+2q}}z_{i,r}\right)^2 - 1\right| \geq \max\{\epsilon, \epsilon^2\}\right) \tag{53}$$

$$\leq 2\exp\left(-\frac{nm}{8}\epsilon^2\right) \tag{54}$$

Therefore,

$$\mathbb{P}\left(\left|\sqrt{\frac{1}{n}\sum_{i=1}^{n}m^{2q}\|z_i\|_2^2} - 1\right| \geq \epsilon\right) \leq 2\exp\left(-\frac{nm}{8}\epsilon^2\right), \tag{55}$$

with probability at least $1 - \delta$, we have

$$\left| m^q \sqrt{\frac{1}{n} \sum_{i=1}^n \|\boldsymbol{z}_i\|_2^2} - 1 \right| \leq \sqrt{\frac{8}{mn} log \frac{2}{\delta}} \tag{56}$$

$$\Rightarrow \left| \sqrt{\frac{1}{n} \sum_{i=1}^n \|\boldsymbol{z}_i\|_2^2} - \frac{1}{m^q} \right| \leq \frac{1}{m^q} \sqrt{\frac{8}{mn} log \frac{2}{\delta}} \tag{57}$$

$\square$

Now, we prove the theorem 5.3. Given the linear model and training dataset $\mathcal{D} = \{(\boldsymbol{x}_i, y_i)\}_{i=1}^n$. Assume that each $w_{i,j} \overset{i.i.d.}{\sim} N(0, \frac{1}{m^{1+2q}}), q > 0$ and $\boldsymbol{a}$ is randomly initialized subject to the constraint $\|\boldsymbol{a}\|_2 = 1$ and fixed during training. Hence, with high probability, we have

$$\dot{\varrho}_{\boldsymbol{a},W} \geq \eta \varrho_{\boldsymbol{a},W} \cdot \left( \frac{\sum_{\tau=1}^t \widetilde{\boldsymbol{x}}(\tau)^T \widetilde{\boldsymbol{x}}(t)}{\|W(t)\|_2^2} \cdot \left(1 - \left(\boldsymbol{v}(t)^T \boldsymbol{a}\right)^2\right) + \mathcal{O}\left(\frac{1}{m^q}\right) \right) \tag{58}$$

*Proof.* We first show that the derivative of co-correlation is

$$\dot{\varrho}_{\boldsymbol{a},W}(t) = \frac{d}{dt} \varrho_{\boldsymbol{a},W}(t) \tag{59}$$

$$= \frac{d}{dt} \frac{\mathbb{E}_{\boldsymbol{x}}[\|\boldsymbol{a}^T W(t)\|_2^2]^{\frac{1}{2}}}{\mathbb{E}_{\boldsymbol{x}}[\|W(t)\|_2^2]^{\frac{1}{2}}} \tag{60}$$

$$= \frac{d}{dt} \frac{\|\boldsymbol{a}^T W(t)\|_2}{\|W(t)\|_2} \tag{61}$$

$$= \frac{\varrho_{\boldsymbol{a},W}(t)}{2} \left( \frac{d\|\boldsymbol{a}^T W(t)\|_2^2/dt}{\|\boldsymbol{a}^T W(t)\|_2^2} - \frac{d\|W(t)\|_2^2/dt}{\|W(t)\|_2^2} \right) \tag{62}$$

$$\geq \frac{\varrho_{\boldsymbol{a},W}(t)}{2\|W(t)\|_2^2} \left( \frac{d\|\boldsymbol{a}^T W(t)\|_2^2}{dt} - \frac{d\|W(t)\|_2^2}{dt} \right), \tag{63}$$

by the law of derivatives for inner product of matrices and the eigenvalue of matrix $W(t)W(t)^T$, we have

$$\frac{d\|\boldsymbol{a}^T W(t)\|_2^2}{dt} = \boldsymbol{a}^T (\dot{W}(t)W(t)^T + W(t)\dot{W}(t)^T)\boldsymbol{a} \tag{64}$$

$$\frac{d\|W(t)\|_2^2}{dt} = \boldsymbol{v(t)}^T (\dot{W}(t)W(t)^T + W(t)\dot{W}(t)^T)\boldsymbol{v}(t), \tag{65}$$

where $\boldsymbol{v}(t)$ is the dominant eigenvector for $W(t)W(t)^T$. Equation (60) is because $J_W(x) = W$ and $\|\boldsymbol{a}\|_2 = 1$ by assumption. Since Equation (60) do not depend on $x$, we can safely drop the Expectation $\mathbb{E}_{\boldsymbol{x}}$ as is shown in Equation (61). Equation (63) is because $\|\boldsymbol{a}^T W\|_2^2 \leq \|\boldsymbol{a}\|_2^2 \|W\|_2^2 = \|W\|_2^2$. By Lemma B.1, the weight matrix $W(t)$ at training step $t$ can be approximated as

$$W(t) = W(0) + \sum_{\tau=1}^t \Delta W(\tau). \tag{66}$$

then

$$\dot{W}(t)W(t)^T = \boldsymbol{a} \otimes \widetilde{\boldsymbol{x}}^T(t)\Big(W(0)^T + \sum_{\tau=1}^{t}\Delta W(\tau)^T\Big) \tag{67}$$

$$= \boldsymbol{a} \otimes \widetilde{\boldsymbol{x}}^T(t)\Big(W(0)^T + \eta\sum_{\tau=1}^{t}\boldsymbol{a}^T \otimes \widetilde{\boldsymbol{x}}(\tau)\Big) \tag{68}$$

$$= \boldsymbol{a} \otimes \widetilde{\boldsymbol{x}}^T(t)\Big(W(0)^T + \eta\boldsymbol{a}^T \otimes \sum_{\tau=1}^{t}\widetilde{\boldsymbol{x}}(\tau)\Big) \tag{69}$$

$$= \boldsymbol{a} \otimes \widetilde{\boldsymbol{x}}^T(t)W(0)^T + \eta\Big(\boldsymbol{a} \otimes \widetilde{\boldsymbol{x}}^T(t)\Big)\Big(\boldsymbol{a}^T \otimes \sum_{\tau=1}^{t}\widetilde{\boldsymbol{x}}(\tau)\Big) \tag{70}$$

$$= \boldsymbol{a}\widetilde{\boldsymbol{x}}^T(t)W(0)^T + \eta\Big(\sum_{\tau=1}^{t}\widetilde{\boldsymbol{x}}(\tau)^T\widetilde{\boldsymbol{x}}(t)\Big)\boldsymbol{a}\boldsymbol{a}^T, \tag{71}$$

similarity,

$$W(t)\dot{W}(t)^T = \Big(W(0) + \sum_{\tau=1}^{t}\Delta W(\tau)\Big)\boldsymbol{a}^T \otimes \widetilde{\boldsymbol{x}}(t) \tag{72}$$

$$= \Big(W(0) + \eta\boldsymbol{a} \otimes \sum_{\tau=1}^{t}\widetilde{\boldsymbol{x}}^T(\tau)\Big)\boldsymbol{a}^T \otimes \widetilde{\boldsymbol{x}}(t) \tag{73}$$

$$= W(0)\widetilde{\boldsymbol{x}}(t)\boldsymbol{a}^T + \eta\Big(\sum_{\tau=1}^{t}\widetilde{\boldsymbol{x}}(\tau)^T\widetilde{\boldsymbol{x}}(t)\Big)\boldsymbol{a}\boldsymbol{a}^T. \tag{74}$$

Hence

$$\frac{1}{2}\Big(\frac{d\|\boldsymbol{a}^T W(t)\|_2^2}{dt} - \frac{d\|W(t)\|_2^2}{dt}\Big) \tag{75}$$

$$= \underbrace{\eta\Big(1 - (\boldsymbol{a}^T\boldsymbol{v}(t))^2\Big)\Big(\sum_{\tau=1}^{t}\widetilde{\boldsymbol{x}}(\tau)^T\widetilde{\boldsymbol{x}}(t)\Big)}_{\text{①}} + \underbrace{\boldsymbol{a}^T W(0)\widetilde{\boldsymbol{x}}(t)\boldsymbol{a}^T\boldsymbol{a} - \boldsymbol{v}(t)^T W(0)\widetilde{\boldsymbol{x}}(t)\boldsymbol{a}^T\boldsymbol{v}(t)}_{\text{②}} \tag{76}$$

① is the main part of our theorem. And for ②, it can be bounded as

$$|②| = \Big|\big(\boldsymbol{a} - \boldsymbol{a}^T\boldsymbol{v}(t)\boldsymbol{v}(t)\big)^T W(0)\widetilde{\boldsymbol{x}}(t)\Big| \tag{77}$$

$$\leq \frac{1}{n}\sum_{i=1}^{n}\Big|\big(y_i - sig(u_i)\big)\big(\boldsymbol{a} - \boldsymbol{a}^T\boldsymbol{v}(t)\boldsymbol{v}(t)\big)^T W(0)\boldsymbol{x}_i\Big| \tag{78}$$

$$\leq \frac{1}{n}\sum_{i=1}^{n}|y_i - sig(u_i)|\underbrace{\sqrt{1 - (\boldsymbol{a}^T\boldsymbol{v}(t))^2}}_{0\leq\ldots\leq 1}\|W(0)\boldsymbol{x}_i\|_2 \tag{79}$$

$$\leq \frac{1}{n}\sum_{i=1}^{n}\|W(0)\boldsymbol{x}_i\|_2 \tag{80}$$

Now since $w_{i,j} \sim N(0, \frac{1}{m^{1+2q}}), q > 0$ and each $w_{i,j}$ is independent with each other, for given $\boldsymbol{x}_i, i = 1, ..., n$, we have

$$W(0)\boldsymbol{x}_i = \begin{pmatrix} \boldsymbol{w}_1(0)^T\boldsymbol{x}_i \\ \vdots \\ \boldsymbol{w}_m(0)^T\boldsymbol{x}_i \end{pmatrix} \sim N\Big(0, \frac{\|\boldsymbol{x}_i\|_2^2}{m^{1+2q}}I_m\Big), \tag{81}$$

where $\|x_i\|_2^2 = 1$ by assumption. By the lemma B.2, with high probability,

$$\frac{1}{n} \sum_{i=1}^{n} \|W(0)x_i\|_2 = \frac{1}{m^q} + \mathcal{O}\left( \sqrt{\frac{8 \log(2n/\delta)}{m^{1+2q}}} \right). \tag{82}$$

Therefore,

$$|②| = \mathcal{O}\left( \frac{1}{m^q} \right) \tag{83}$$

with high probability, and it comes to our conclusion. □

## C  The Proof of Theorem 5.5

To prove the Theorem 5.5, the following lemma is crucial.

**Lemma C.1** (Dynamic of Weights for A Nonlinear Model for Initial Steps). *Given gradient descent to optimize the weights, the dynamic of the weights at the initial step $t$ for the non-linear model is*

$$\dot{W}(t) = \begin{pmatrix} a_1 \widetilde{x}_1^T(t) \\ \vdots \\ a_m \widetilde{x}_m^T(t) \end{pmatrix}, \tag{84}$$

*where*

$$\widetilde{x}_r(t) = a_r \frac{1}{n} \sum_{i=1}^{n} \left( y_i - sig(u_i(t)) \right) \sigma'(w_r^T(t)x_i)x_i, \quad r \in [m]. \tag{85}$$

*Proof.* Similar to the linear case, the update for the $r_{th}$ row for the weight matrix is

$$\Delta w_r = -\frac{\eta}{n} \sum_{i=1}^{n} \frac{\partial l}{\partial u_i} \frac{\partial f(W, x_i)}{\partial w_r} \tag{86}$$

$$= \eta a_r \frac{1}{n} \sum_{i=1}^{n} \left( y_i - sig(u_i) \right) \sigma'(w_r^T x_i)x_i. \tag{87}$$

Hence that the update of the weight matrix $W(t)$ is

$$\Delta W = \begin{pmatrix} \Delta w_1^T \\ \vdots \\ \Delta w_m^T \end{pmatrix} \tag{88}$$

$$= \eta \begin{pmatrix} a_1 \frac{1}{n} \sum_{i=1}^{n} \left( y_i - sig(u_i) \right) \sigma'(w_1^T x_i)x_i^T \\ \vdots \\ a_m \frac{1}{n} \sum_{i=1}^{n} \left( y_i - sig(u_i) \right) \sigma'(w_m^T x_i)x_i^T \end{pmatrix} \tag{89}$$

$$= \eta \begin{pmatrix} a_1 \widetilde{x}_1^T \\ \vdots \\ a_m \widetilde{x}_m^T \end{pmatrix}. \tag{90}$$

Hence our conclusion. □

**Lemma C.2** (Concentration for Weighted Sum and Square Root Scalar for Bounded Variables). *Suppose $X_r - \mu \in [l, u], \forall i \in [m]$ is independent identically distributed r.v. with mean 0. Given a vector $v \in \mathbb{R}^m, \|v\|_2 = 1$, and $a$ such that $a_r \in \{-1, 1\}, r \in [m]$ we have concentration inequality*

$$\mathbb{P}\left( \left| \frac{1}{\sqrt{m}} \sum_{r=1}^{m} a_r v_r X_r - \frac{1}{\sqrt{m}} a^T v \mu \right| \geq \epsilon \right) \leq 2 \exp \left\{ -\frac{m\epsilon^2}{2(u-l)^2} \right\}. \tag{91}$$

*Proof.* We first show the moment generation function of the r.v. we would like to estimate.

$$\mathbb{E}\left[\exp\left\{t\left(\frac{1}{\sqrt{m}}\sum_{r=1}^{m}a_r v_r X_r - \frac{1}{\sqrt{m}}\boldsymbol{a}^T\boldsymbol{v}\mu\right)\right\}\right] \leq \exp\left\{\frac{t^2}{2m}(u-l)^2\right\}. \tag{92}$$

Now, we prove this inequality.

$$\mathbb{E}\left[\exp\left\{t\left(\frac{1}{\sqrt{m}}\sum_{r=1}^{m}a_r v_r X_r - \frac{1}{\sqrt{m}}\boldsymbol{a}^T\boldsymbol{v}\mu\right)\right\}\right] = \mathbb{E}\left[\exp\left\{\frac{t}{\sqrt{m}}\left(\sum_{r=1}^{m}a_r v_r X_r - \sum_{r=1}^{m}a_r v_r\mu\right)\right\}\right] \tag{93}$$

$$= \prod_{r=1}^{m}\mathbb{E}\left[\exp\left\{\frac{ta_r v_r}{\sqrt{m}}\left(X_r - \mu\right)\right\}\right] \tag{94}$$

$$\leq \prod_{r=1}^{m}\exp\left\{\frac{v_r^2 t^2}{2m}(u-l)^2\right\} \tag{95}$$

$$= \exp\left\{\frac{t^2}{2m}(u-l)^2\sum_{r=1}^{m}v_r^2\right\} \tag{96}$$

$$= \exp\left\{\frac{t^2}{2m}(u-l)^2\right\}, \tag{97}$$

where $t \geq 0$. Hence by Chernoff's bound, we have that

$$\mathbb{P}\left(\left|\frac{1}{\sqrt{m}}\sum_{r=1}^{m}a_r v_r X_r - \frac{1}{\sqrt{m}}\boldsymbol{a}^T\boldsymbol{v}\mu\right| \geq \epsilon\right) \leq 2\inf_{t\geq 0}\frac{\exp\left\{\frac{t^2}{2m}(u-l)^2\right\}}{\exp\{t\epsilon\}} \tag{98}$$

$$= 2\exp\left\{-\frac{m\epsilon^2}{2(u-l)^2}\right\} \tag{99}$$

$\square$

Here, we prove the theorem 5.5. Given a shallow neural network, with high probability we have

$$\dot{\varrho}_{\boldsymbol{a},\sigma\circ W}(t) = \frac{\eta\varrho_{\boldsymbol{a},\sigma\circ W}(t)}{\mathbb{E}_{\boldsymbol{x}}\|D(t)W(t)\|_2^2}\left[\sum_{\tau=1}^{t}(1 - \boldsymbol{a}^T\boldsymbol{v}(\tau)\boldsymbol{a}^T\boldsymbol{v}(t))\mathbb{E}_{\boldsymbol{x}}\left[\widetilde{\boldsymbol{x}}^*(\tau)^T\widetilde{\boldsymbol{x}}^*(t)\right]\right. \tag{100}$$

$$\left. + \max\left\{\mathcal{O}\left(\frac{1}{\sqrt{m}}\right), \mathcal{O}\left(\frac{1}{m^q}\right)\right\}\right] \tag{101}$$

*Proof.* Because of activation function depends on the input $\boldsymbol{x}$, we cannot drop the expectation $\mathbb{E}_{\boldsymbol{x}}$. Hence, we have

$$\dot{\varrho}_{\boldsymbol{a},W}(t) = \frac{d}{dt}\varrho_{\boldsymbol{a},W}(t) \tag{102}$$

$$= \frac{d}{dt}\frac{\mathbb{E}_{\boldsymbol{x}}[\|\boldsymbol{a}^T D(t)W(t)\|_2^2]^{\frac{1}{2}}}{\mathbb{E}_{\boldsymbol{x}}[\|D(t)W(t)\|_2^2]^{\frac{1}{2}}} \tag{103}$$

$$= \frac{\varrho_{\boldsymbol{a},W}(t)}{2}\left(\frac{\mathbb{E}_{\boldsymbol{x}}d\|\boldsymbol{a}^T D(t)W(t)\|_2^2/dt}{\mathbb{E}_{\boldsymbol{x}}\|\boldsymbol{a}^T D(t)W(t)\|_2^2} - \frac{\mathbb{E}_{\boldsymbol{x}}d\|D(t)W(t)\|_2^2/dt}{\mathbb{E}_{\boldsymbol{x}}\|D(t)W(t)\|_2^2}\right) \tag{104}$$

$$\geq \frac{\varrho_{\boldsymbol{a},W}(t)}{2\mathbb{E}_{\boldsymbol{x}}\|D(t)W(t)\|_2^2}\mathbb{E}_{\boldsymbol{x}}\left(\frac{d\|\boldsymbol{a}^T D(t)W(t)\|_2^2}{dt} - \frac{d\|D(t)W(t)\|_2^2}{dt}\right), \tag{105}$$

where $D(t) = diag(\sigma'(\boldsymbol{w}_1^T\boldsymbol{x}),...,\sigma'(\boldsymbol{w}_m^T\boldsymbol{x}))$. And for ReLu activation function, $\sigma'(\boldsymbol{w}_m^T\boldsymbol{x}) = \mathbf{1}_{\{\boldsymbol{w}_m^T\boldsymbol{x}\geq 0\}}$. We assume that the activation does not change at an infinitely small change of $t$, implying $\dot{D}(t) = \mathbf{0}$. Hence, we have

$$\frac{d\|\boldsymbol{a}^T D(t)W(t)\|_2^2}{dt} = \boldsymbol{a}^T \Big( \dot{D}(t)W(t)W(t)^T D(t) + D(t)W(t)W(t)^T \dot{D}(t)$$

$$+ D(t)\dot{W}(t)W(t)^T D(t) + D(t)W(t)\dot{W}(t)^T D(t) \Big)\boldsymbol{a}$$

$$= \boldsymbol{a}^T \Big( D(t)\dot{W}(t)W(t)^T D(t) + D(t)W(t)\dot{W}(t)^T D(t) \Big)\boldsymbol{a}$$

$$= 2\,\boldsymbol{a}^T D(t)\eta \sum_{\tau=1}^{t} \underbrace{\begin{pmatrix} a_1\widetilde{\boldsymbol{x}}_1^T(\tau) \\ \vdots \\ a_m\widetilde{\boldsymbol{x}}_m^T(\tau) \end{pmatrix} \begin{pmatrix} a_1\widetilde{\boldsymbol{x}}_1(t) & \cdots & a_m\widetilde{\boldsymbol{x}}_m(t) \end{pmatrix} D(t)\boldsymbol{a}}_{①}$$

$$+ 2\,\boldsymbol{a}^T D(t) \underbrace{\begin{pmatrix} a_1\widetilde{\boldsymbol{x}}_1^T(t) \\ \vdots \\ a_m\widetilde{\boldsymbol{x}}_m^T(t) \end{pmatrix} W(0)^T D(t)\boldsymbol{a}}_{②}\,.$$

And similarly,

$$\frac{d\|D(t)W(t)\|_2^2}{dt} = \boldsymbol{v}^T(t) \Big( D(t)\dot{W}(t)W(t)^T D(t) + D(t)W(t)\dot{W}(t)^T D(t) \Big)\boldsymbol{v}(t)$$

$$= 2\,\boldsymbol{v}^T(t)D(t)\eta \sum_{\tau=1}^{t} \underbrace{\begin{pmatrix} a_1\widetilde{\boldsymbol{x}}_1^T(\tau) \\ \vdots \\ a_m\widetilde{\boldsymbol{x}}_m^T(\tau) \end{pmatrix} \begin{pmatrix} a_1\widetilde{\boldsymbol{x}}_1(t) & \cdots & a_m\widetilde{\boldsymbol{x}}_m(t) \end{pmatrix} D(t)\boldsymbol{v}(t)}_{①'}$$

$$+ 2\,\boldsymbol{v}^T(t)D(t) \underbrace{\begin{pmatrix} a_1\widetilde{\boldsymbol{x}}_1^T(t) \\ \vdots \\ a_m\widetilde{\boldsymbol{x}}_m^T(t) \end{pmatrix} W(0)^T D(t)\boldsymbol{v}(t)}_{②'}\,.$$

① can be

$$① = \eta \sum_{\tau=1}^{t} \left( \sum_{r=1}^{m} a_r^2 \sigma'(\boldsymbol{w}_r(\tau)^T \boldsymbol{x})\widetilde{\boldsymbol{x}}_r^T(\tau) \right)\left( \sum_{r=1}^{m} a_r^2 \sigma'(\boldsymbol{w}_r(t)^T \boldsymbol{x})\widetilde{\boldsymbol{x}}_r(t) \right) \tag{106}$$

$$= \eta \sum_{\tau=1}^{t} \left[ \frac{1}{n} \sum_{i=1}^{n} \sum_{r=1}^{m} \left[ a_r^2 \sigma'(\boldsymbol{w}_r(\tau)^T \boldsymbol{x})\sigma'(\boldsymbol{w}_r(\tau)^T \boldsymbol{x}_i) \right](y - sig(u_i))\boldsymbol{x}_i^T \right] \tag{107}$$

$$\cdot \left[ \frac{1}{n} \sum_{i=1}^{n} \sum_{r=1}^{m} \left[ a_r^2 \sigma'(\boldsymbol{w}_r(t)^T \boldsymbol{x})\sigma'(\boldsymbol{w}_r(t)^T \boldsymbol{x}_i) \right](y - sig(u_i))\boldsymbol{x}_i \right]. \tag{108}$$

Since $a_r \in \{-\frac{1}{\sqrt{m}}, \frac{1}{\sqrt{m}}\}$ and the derivative activation function $\sigma'$ is bounded by $M$ by assumption, we have that

$$\mathbb{P}_{W(0)}\left( \left| \sum_{r=1}^{m} a_r^2 \sigma'(\boldsymbol{w}_r(t)^T \boldsymbol{x})\sigma'(\boldsymbol{w}_r(t)^T \boldsymbol{x}_i) - \alpha_i(t;\boldsymbol{x}) \right| \geq \epsilon \right) \tag{109}$$

$$= \mathbb{P}_{W(0)}\left( \left| \frac{1}{m} \sum_{r=1}^{m} \sigma'(\boldsymbol{w}_r(t)^T \boldsymbol{x})\sigma'(\boldsymbol{w}_r(t)^T \boldsymbol{x}_i) - \alpha_i(t;\boldsymbol{x}) \right| \geq \epsilon \right) \leq 2e^{-\frac{m\epsilon^2}{2M^4}}, \tag{110}$$

where

$$\alpha_i(t, \boldsymbol{x}) = \mathbb{E}_{W(0)}\Big[\sigma'(\boldsymbol{w}(t)^T\boldsymbol{x})\sigma'(\boldsymbol{w}(t)^T\boldsymbol{x}_i)\Big]. \tag{111}$$

Equivalently, with probability at least $1 - \delta$, we have

$$\Big|\sum_{r=1}^{m} a_r^2 \sigma'(\boldsymbol{w}_r(t)^T\boldsymbol{x})\sigma'(\boldsymbol{w}_r(t)^T\boldsymbol{x}_i) - \alpha_i(t; \boldsymbol{x})\Big| \leq \sqrt{\frac{2M^4}{m}log\frac{2}{\delta}} \tag{112}$$

Let

$$\widetilde{\boldsymbol{x}}^*(t) \triangleq \frac{1}{n}\sum_{i=1}^{n} \alpha_i(t, \boldsymbol{x})\big(y_i - sig(u_i(t))\big)\boldsymbol{x}_i \tag{113}$$

$$\widetilde{\boldsymbol{x}}(t) \triangleq \frac{1}{n}\sum_{i=1}^{n} \big(y_i - sig(u_i(t))\big)\boldsymbol{x}_i \tag{114}$$

Hence, with probability at least $1 - \delta$,

$$① = \eta\sum_{\tau=1}^{t}\Big[\frac{1}{n}\sum_{i=1}^{n}\alpha_i(\tau, \boldsymbol{x})\big(y_i - sig(u_i(\tau))\big)\boldsymbol{x}_i^T + \mathcal{O}\Big(\sqrt{\frac{1}{m}\log\frac{2}{\delta}}\Big)\widetilde{\boldsymbol{x}}(\tau)^T\Big] \tag{115}$$

$$\cdot\Big[\frac{1}{n}\sum_{i=1}^{n}\alpha_i(t, \boldsymbol{x})\big(y_i - sig(u_i(t))\big)\boldsymbol{x}_i + \mathcal{O}\Big(\sqrt{\frac{1}{m}\log\frac{2}{\delta}}\Big)\widetilde{\boldsymbol{x}}(t)\Big] \tag{116}$$

$$= \eta\sum_{\tau=1}^{t}\Big[\widetilde{\boldsymbol{x}}^*(\tau)^T + \mathcal{O}\Big(\sqrt{\frac{1}{m}\log\frac{2}{\delta}}\Big)\widetilde{\boldsymbol{x}}(\tau)^T\Big]\cdot\Big[\widetilde{\boldsymbol{x}}^*(t) + \mathcal{O}\Big(\sqrt{\frac{1}{m}\log\frac{2}{\delta}}\Big)\widetilde{\boldsymbol{x}}(t)\Big] \tag{117}$$

$$= \eta\sum_{\tau=1}^{t}\widetilde{\boldsymbol{x}}^*(\tau)^T\widetilde{\boldsymbol{x}}^*(t) + \mathcal{O}\Big(\sqrt{\frac{1}{m}\log\frac{2}{\delta}}\Big)\widetilde{\boldsymbol{x}}(\tau)^T\widetilde{\boldsymbol{x}}^*(t) \tag{118}$$

$$+ \mathcal{O}\Big(\sqrt{\frac{1}{m}\log\frac{2}{\delta}}\Big)\widetilde{\boldsymbol{x}}^*(\tau)^T\widetilde{\boldsymbol{x}}(t) + \mathcal{O}\Big(\frac{1}{m}\log\frac{2}{\delta}\Big)\widetilde{\boldsymbol{x}}(t)^T\widetilde{\boldsymbol{x}}(t). \tag{119}$$

Since

$$\|\widetilde{\boldsymbol{x}}^*(t)\|_2 \leq \frac{1}{n}\Big\|\begin{pmatrix}\alpha_1(\tau, \boldsymbol{x})\big(y_1 - sig(u_1(t))\big)\\ \vdots \\ \alpha_n(\tau, \boldsymbol{x})\big(y_n - sig(u_n(t))\big)\end{pmatrix}\Big\|_2\Big\|\begin{pmatrix}\boldsymbol{x}_1^T\\ \vdots \\ \boldsymbol{x}_n^T\end{pmatrix}\Big\|_2 \leq M^2 \tag{120}$$

$$\|\widetilde{\boldsymbol{x}}(t)\|_2 \leq \frac{1}{n}\Big\|\begin{pmatrix}y_1 - sig(u_1(t))\\ \vdots \\ y_n - sig(u_n(t))\end{pmatrix}\Big\|_2\Big\|\begin{pmatrix}\boldsymbol{x}_1^T\\ \vdots \\ \boldsymbol{x}_n^T\end{pmatrix}\Big\|_2 \leq 1, \tag{121}$$

we have that for at least $1 - \delta$,

$$① = \eta\sum_{\tau=1}^{t}\widetilde{\boldsymbol{x}}^*(\tau)^T\widetilde{\boldsymbol{x}}^*(t) + \mathcal{O}\Big(\sqrt{\frac{1}{m}\log\frac{2}{\delta}}\Big) \tag{122}$$

Similarly,

$$①' = \eta\sum_{\tau=1}^{t}\Big[\frac{1}{n}\sum_{i=1}^{n}\sum_{r=1}^{m}\Big[v_r(t)a_r\sigma'(\boldsymbol{w}_r(\tau)^T\boldsymbol{x})\sigma'(\boldsymbol{w}_r(\tau)^T\boldsymbol{x}_i)\Big](y_i - sig(u_i))\boldsymbol{x}_i^T\Big] \tag{123}$$

$$\cdot\Big[\frac{1}{n}\sum_{i=1}^{n}\sum_{r=1}^{m}\Big[v_r(t)a_r\sigma'(\boldsymbol{w}_r(t)^T\boldsymbol{x})\sigma'(\boldsymbol{w}_r(t)^T\boldsymbol{x}_i)\Big](y_i - sig(u_i))\boldsymbol{x}_i\Big], \tag{124}$$

Let $\boldsymbol{a}^* = \sqrt{m}\boldsymbol{a} \in \{-1, 1\}^m$ and by the conclusion of lemma C.2,

$$\mathbb{P}_{W(0)}\left(\left|\sum_{r=1}^{m} v_r(t)a_r\sigma'(\boldsymbol{w}_r(t)^T\boldsymbol{x})\sigma'(\boldsymbol{w}_r(t)^T\boldsymbol{x}_i) - \sum_{r=1}^{m} v_r(t)a_r\alpha_i(t,\boldsymbol{x})\right| \geq \epsilon\right) \tag{125}$$

$$= \mathbb{P}_{W(0)}\left(\left|\frac{1}{\sqrt{m}}\sum_{r=1}^{m} a_r^* v_r(t)\sigma'(\boldsymbol{w}_r(t)^T\boldsymbol{x})\sigma'(\boldsymbol{w}_r(t)^T\boldsymbol{x}_i) - \boldsymbol{a}^T\boldsymbol{v}(t)\alpha_i(t;\boldsymbol{x})\right| \geq \epsilon\right) \tag{126}$$

$$\leq 2\exp\left\{-\frac{m\epsilon^2}{2M^4}\right\}. \tag{127}$$

Equivalently, with probability at least $1 - \delta$,

$$\left|\frac{1}{\sqrt{m}}\sum_{r=1}^{m} a_r^* v_r(t)\sigma'(\boldsymbol{w}_r(t)^T\boldsymbol{x})\sigma'(\boldsymbol{w}_r(t)^T\boldsymbol{x}_i) - \boldsymbol{a}^T\boldsymbol{v}(t)\alpha_i(t;\boldsymbol{x})\right| \leq \sqrt{\frac{2M^4}{m}\log\frac{2}{\delta}}. \tag{128}$$

Hence,

$$①' = \eta\sum_{\tau=1}^{t}\left[\frac{1}{n}\sum_{i=1}^{n}\boldsymbol{a}^T\boldsymbol{v}(\tau)\alpha_i(\tau,\boldsymbol{x})\big(y_i - sig(u_i(\tau))\big)\boldsymbol{x}_i^T + \mathcal{O}\left(\sqrt{\frac{1}{m}\log\frac{2}{\delta}}\right)\widetilde{\boldsymbol{x}}(\tau)^T\right] \tag{129}$$

$$\cdot\left[\frac{1}{n}\sum_{i=1}^{n}\boldsymbol{a}^T\boldsymbol{v}(t)\alpha_i(t,\boldsymbol{x})\big(y_i - sig(u_i(t))\big)\boldsymbol{x}_i + \mathcal{O}\left(\sqrt{\frac{1}{m}\log\frac{2}{\delta}}\right)\widetilde{\boldsymbol{x}}(t)\right] \tag{130}$$

$$= \eta\sum_{\tau=1}^{t}\left[\boldsymbol{a}^T\boldsymbol{v}(\tau)\widetilde{\boldsymbol{x}}^*(\tau)^T + \mathcal{O}\left(\sqrt{\frac{1}{m}\log\frac{2}{\delta}}\right)\widetilde{\boldsymbol{x}}(\tau)^T\right]\cdot\left[\boldsymbol{a}^T\boldsymbol{v}(t)\widetilde{\boldsymbol{x}}^*(t) + \mathcal{O}\left(\sqrt{\frac{1}{m}\log\frac{2}{\delta}}\right)\widetilde{\boldsymbol{x}}(t)\right] \tag{131}$$

$$= \eta\sum_{\tau=1}^{t}\boldsymbol{a}^T\boldsymbol{v}(\tau)\boldsymbol{a}^T\boldsymbol{v}(t)\widetilde{\boldsymbol{x}}^*(\tau)^T\widetilde{\boldsymbol{x}}^*(t) \tag{132}$$

$$+ \mathcal{O}\left(\sqrt{\frac{1}{m}\log\frac{2}{\delta}}\right)\widetilde{\boldsymbol{x}}(\tau)^T\widetilde{\boldsymbol{x}}^*(t) + \mathcal{O}\left(\sqrt{\frac{1}{m}\log\frac{2}{\delta}}\right)\widetilde{\boldsymbol{x}}^*(\tau)^T\widetilde{\boldsymbol{x}}(t) + \mathcal{O}\left(\frac{1}{m}\log\frac{2}{\delta}\right)\widetilde{\boldsymbol{x}}(t)^T\widetilde{\boldsymbol{x}}(t). \tag{133}$$

We have that for at least $1 - \delta$,

$$①' = \eta\sum_{\tau=1}^{t}\boldsymbol{a}^T\boldsymbol{v}(\tau)\boldsymbol{a}^T\boldsymbol{v}(t)\widetilde{\boldsymbol{x}}^*(\tau)^T\widetilde{\boldsymbol{x}}^*(t) + \mathcal{O}\left(\sqrt{\frac{1}{m}\log\frac{2}{\delta}}\right) \tag{134}$$

For $②$, we have

$$|②| = \left|\boldsymbol{a}^T D(t)\begin{pmatrix} a_1\widetilde{\boldsymbol{x}}_1^T(t) \\ \vdots \\ a_m\widetilde{\boldsymbol{x}}_m^T(t) \end{pmatrix} W(0)^T D(t)\boldsymbol{a}\right| \tag{135}$$

$$\leq \|\boldsymbol{a}^T D(t)\|_2^2\big\| \big(a_1 W(0)\widetilde{\boldsymbol{x}}_1(t) \quad \cdots \quad a_m W(0)\widetilde{\boldsymbol{x}}_m(t)\big)\big\|_2 \tag{136}$$

$$\leq M^2\big\| \big(a_1 W(0)\widetilde{\boldsymbol{x}}_1(t) \quad \cdots \quad a_m W(0)\widetilde{\boldsymbol{x}}_m(t)\big)\big\|_F \tag{137}$$

$$\leq M^2\sqrt{\sum_{r=1}^{m} a_r^2\|W(0)\widetilde{\boldsymbol{x}}_r\|_2^2}, \tag{138}$$

where

$$\|W(0)\widetilde{\boldsymbol{x}}_r\|_2^2 = \left\|\frac{1}{n}\sum_{i=1}^{n}\sigma'(\boldsymbol{w}_r(0)^T\boldsymbol{x}_i)(y_i - sig(u_i))W(0)\boldsymbol{x}_i\right\|_2^2 \tag{139}$$

$$\leq \frac{1}{n^2}\left\|\begin{pmatrix}\sigma'(\boldsymbol{w}_r(0)^T\boldsymbol{x}_1)(y_1 - sig(u_1))\\ \vdots\\ \sigma'(\boldsymbol{w}_r(0)^T\boldsymbol{x}_n)(y_n - sig(u_n))\end{pmatrix}\right\|_2^2\left\|\begin{pmatrix}W(0)\boldsymbol{x}_1 & \cdots & W(0)\boldsymbol{x}_n\end{pmatrix}\right\|_2^2 \tag{140}$$

$$\leq \frac{1}{n^2}\sum_{i=1}^{n}\underbrace{\sigma'(\boldsymbol{w}_r(0)^T\boldsymbol{x}_i)^2(y_i - sig(u_i))^2}_{0\leq\dots\leq M^2}\left\|\begin{pmatrix}W(0)\boldsymbol{x}_1 & \cdots & W(0)\boldsymbol{x}_n\end{pmatrix}\right\|_F^2 \tag{141}$$

$$\leq \frac{M^2}{n}\sum_{i=1}^{n}\|W(0)\boldsymbol{x}_i\|_2^2 \tag{142}$$

Similar to the proof of the linear case, and accords to Lemma B.2, we have that with probability at least $1 - \delta$,

$$\sqrt{\frac{1}{n}\sum_{i=1}^{n}\|W(0)\boldsymbol{x}_i\|_2^2} = \frac{1}{m^q} + \mathcal{O}\left(\sqrt{\frac{8\log(2/\delta)}{m^{1+2q}n}}\right). \tag{143}$$

Therefore, with high probability,

$$|\textcircled{2}| \leq M^4\sqrt{\sum_{r=1}^{m}a_r^2\frac{1}{n}\sum_{i=1}^{n}\|W(0)\boldsymbol{x}_i\|_2^2} \tag{144}$$

$$= M^4\sqrt{\frac{1}{n}\sum_{i=1}^{n}\|W(0)\boldsymbol{x}_i\|_2^2} \tag{145}$$

$$= \mathcal{O}\left(\frac{1}{m^q}\right) \tag{146}$$

Follow the exact same procedure. With high probability, we have

$$|\textcircled{2}'| = \left|\boldsymbol{v}^T(t)D(t)\begin{pmatrix}a_1\widetilde{\boldsymbol{x}}_1^T(t)\\ \vdots\\ a_m\widetilde{\boldsymbol{x}}_m^T(t)\end{pmatrix}W(0)^TD(t)\boldsymbol{v}(t)\right| \tag{147}$$

$$\leq \|\boldsymbol{v}(t)^TD(t)\|_2^2\left\|\begin{pmatrix}a_1W(0)\widetilde{\boldsymbol{x}}_1(t) & \cdots & a_mW(0)\widetilde{\boldsymbol{x}}_m(t)\end{pmatrix}\right\|_2 \tag{148}$$

$$\leq M^2\sqrt{\sum_{r=1}^{m}a_r^2\|W(0)\widetilde{\boldsymbol{x}}_r\|_2^2} \tag{149}$$

$$= \mathcal{O}\left(\frac{1}{m^q}\right) \tag{150}$$

We put all the information together, with high probability, we have

$$\dot{\varrho}_{\boldsymbol{a},\sigma\circ W}(t) \geq \frac{\varrho_{\boldsymbol{a},\sigma\circ W}(t)}{\mathbb{E}_{\boldsymbol{x}}\|D(t)W(t)\|_2^2}\mathbb{E}_{\boldsymbol{x}}\left(\textcircled{1} - \textcircled{1}' + \textcircled{2} - \textcircled{2}'\right) \tag{151}$$

$$\geq \frac{\varrho_{\boldsymbol{a},\sigma\circ W}(t)}{\mathbb{E}_{\boldsymbol{x}}\|D(t)W(t)\|_2^2}\left[\eta\sum_{\tau=1}^{t}(1 - \boldsymbol{a}^T\boldsymbol{v}(\tau)\boldsymbol{a}^T\boldsymbol{v}(t))\mathbb{E}_{\boldsymbol{x}}\left[\widetilde{\boldsymbol{x}}^*(\tau)^T\widetilde{\boldsymbol{x}}^*(t)\right]\right. \tag{152}$$

$$\left. + \max\left\{\mathcal{O}\left(\frac{1}{m^{\frac{1}{2}}}\right), \mathcal{O}\left(\frac{1}{m^q}\right)\right\}\right] \tag{153}$$

$\square$

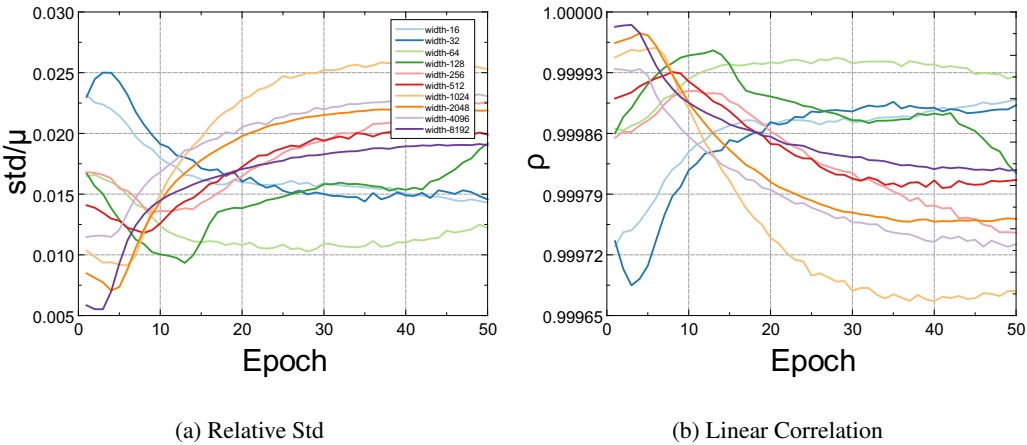

(a) Relative Std           (b) Linear Correlation

Figure 5: Linear Correlation & Relative Std. The linear correlation and the standard deviation over the mean are given for all MLPs with the initialization parameter $q = 0.25$.

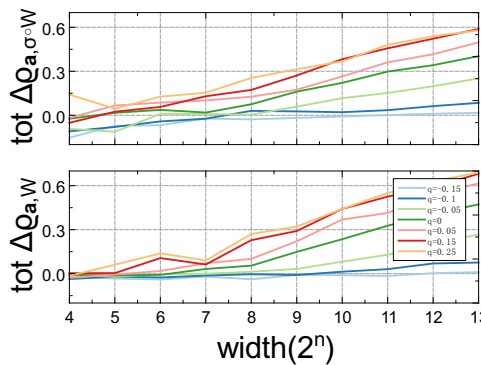

Figure 6: The accumulation of co-correlation under different width. X-axis denotes the width of the neural network, and the line in different color shows the result under different setting of weight initialization. The lower plot is the result for linear model and the result for shallow ReLU is at the top.

## D  Extra Experiment Results

Figure 5 summarizes the statistics of linear correlation and relative variability, illustrating that co-correlation is the dominant factor in the decomposition outlined in Theorem 4.5.

Figure 6 shows the accumulated change in co-correlation during training across varying network widths. Each line, represented by different colours, corresponds to a unique weight initialization setting of $q$. A clear inference from the figure is the positive relationship between the accumulated co-correlation change and network width under the same $q$—the wider the network, the greater the accumulated co-correlation change. Moreover, as we shift the weight initialization setting from $q = -0.15$ to $q = 0.25$, the accumulation of co-correlation also increases. These observations align with our Theorem 5.3 and 5.5, indicating that the increase in co-correlation is inversely proportional to $\|W(t)\|_2$.

# E   Algorithm

We use the *Power Iteration algorithm* [10] along with *Functorch* [17] to compute co-correlation and other relevant statistics. This combination assists in approximating the $L_2$-norm of the Jacobian for the layers under consideration. The algorithm to estimate the related statistics including co-correlation are outlined in Algorithm 1. The power iteration used to obtain the $L_2$-norm of the Jacobian is summarized in Algorithm 2.

---

**Algorithm 1** Estimation of Related Statistics

---

**Input**: Input dataset and modules $\phi, \varphi$ for the given blocks in sequential $\{\boldsymbol{x}_i, i = 1, .., N\}$
**Output**: Approximation of co-correlations $\widehat{\varrho}_{\phi,\varphi}$, linear correlation $\widehat{\rho}_{\phi,\varphi}$, mean $\widehat{\mu}_\phi, \widehat{\mu}_\varphi$ and variance $\sigma_\phi, \sigma_\varphi$ of the Jacobian of the module w.r.t. their inputs

1: Computation of Jacobin of $\|J_\varphi(\boldsymbol{x}_i)\|_2, \|J_\phi(\varphi(\boldsymbol{x}_i))\|_2$ for $\{\boldsymbol{x}_i, i \in [n]\}$ w.r.t. their inputs

2: $\widehat{\varrho}_{\phi,\varphi} \leftarrow \dfrac{\frac{1}{N}\sum_{i=1}^{N}\|J_\phi(\varphi(\boldsymbol{x}_i))\cdot J_\varphi(\boldsymbol{x}_i)\|_2}{\frac{1}{N}\sum_{i=1}^{N}\|J_\phi(\varphi(\boldsymbol{x}_i))\|_2\|J_\varphi(\boldsymbol{x}_i)\|_2}$

3: $\widehat{\rho}_{\phi,\varphi} \leftarrow \dfrac{\frac{1}{N}\sum_{i=1}^{N}\|J_\phi(\varphi(\boldsymbol{x}_i))\|_2\|J_\varphi(\boldsymbol{x}_i)\|_2}{\left(\frac{1}{N}\sum_{i=1}^{N}\|J_\phi(\varphi(\boldsymbol{x}_i))\|_2^2\right)^{\frac{1}{2}}\left(\frac{1}{N}\sum_{i=1}^{N}\|J_\varphi(\boldsymbol{x}_i)\|_2^2\right)^{\frac{1}{2}}}$

4: $\widehat{\mu}_\phi \leftarrow \frac{1}{N}\sum_{i=1}^{N}\|J_\phi(\varphi(\boldsymbol{x}_i))\|_2\|J_\varphi(\boldsymbol{x}_i)\|_2$

5: $\widehat{\sigma}_\phi \leftarrow \left[\frac{1}{N-1}\sum_{i=1}^{N}(\|J_\phi(\varphi(\boldsymbol{x}_i))\|_2\|J_\varphi(\boldsymbol{x}_i)\|_2 - \hat{\mu}_\phi)^2\right]^{\frac{1}{2}}$

---

**Algorithm 2** Power Iteration Based Computation

---

**Input**: Input dataset and module for the given sequential blocks $\{\boldsymbol{x}_i, i = 1, .., N\}, \phi, \varphi$
**Output**: Approximation of co-correlations $\varrho_{\phi,\varphi}$, linear correlation $\rho_{\phi,\varphi}$, mean $\mu_\phi, \mu_\varphi$ and variance $\sigma_\phi, \sigma_\varphi$ of the Jacobian of the module w.r.t. their inputs

1: COMPUTATION OF $\|J_\varphi(\boldsymbol{x}_i)\|_2$ FOR $\{\boldsymbol{x}_i, i \in [n]\}$
2: **while** $i \leq N_b$ **do**
3:     **sample randomly** $\boldsymbol{u} \sim U([0,1])$
4:     $\boldsymbol{u} \leftarrow \boldsymbol{u}/\|\boldsymbol{u}\|_2$
5:     **generate jacobian-vector product function** $jvp(\cdot; \boldsymbol{x}_i)$
6:     **generating vector-jacobian product function** $vjp(\cdot; \boldsymbol{x}_i)$
7:     $\|J_\varphi(\boldsymbol{x}_i)\|_2^{old} \leftarrow None$
8:     **while** $err \geq 10^{-6}$ **do**
9:         $\boldsymbol{v} \leftarrow jvp(\boldsymbol{u}; \boldsymbol{x}_i)$
10:         $\boldsymbol{u} \leftarrow vjp(\boldsymbol{v}; \boldsymbol{x}_i)$
11:         $\boldsymbol{u} \leftarrow \boldsymbol{u}/\|\boldsymbol{u}\|_2$
12:         $\boldsymbol{v} \leftarrow \boldsymbol{v}/\|\boldsymbol{v}\|_2$
13:         $\|J_\varphi(\boldsymbol{x}_i)\|_2^{new} \leftarrow \frac{\boldsymbol{u}/\|\boldsymbol{u}\|_2}{\boldsymbol{v}/\|\boldsymbol{v}\|_2}$
14:         **if** $\|J_\varphi(\boldsymbol{x}_i)\|_2^{old} \neq None$ **then**
15:             $err = \left|\frac{\|J_\varphi(\boldsymbol{x}_i)\|_2^{new} - \|J_\varphi(\boldsymbol{x}_i)\|_2^{old}}{\|J_\varphi(\boldsymbol{x}_i)\|_2^{old}}\right|$
16:         **end if**
17:         $\|J_\varphi(\boldsymbol{x}_i)\|_2^{old} = \|J_\varphi(\boldsymbol{x}_i)\|_2^{new}$
18:     **end while**
19: **end while**

---

