# OpenReview forum: "The Implicit Bias of Gradient Descent toward Collaboration between Layers: A Dynamic Analysis of Multilayer Perceptions"
_NeurIPS.cc/2024/Conference — NeurIPS 2024 poster_

### Official Review · Reviewer_DyJc · 2024-07-12

**Soundness:** 3
**Presentation:** 2
**Contribution:** 3
**Rating:** 5
**Confidence:** 3

**Summary:**

In this work, the authors study the difference between underparameterised and overparameterised networks in terms of the collaboration between consecutive layers. They find that under-parameterized networks tend to foster co-correlation among layers to improve performance, whereas performance of over-parameterized networks can improve without relying on this co-correlation. They relate the co-correlation to adversarial robustness.

**Strengths:**

This paper adresses an important open question in deep learning: how should we relate the concepts of generalization, adversarial robustness and the network archictecture and size? The paper adresses this question from an original perspective.

The literature study is quite extensive and motivates the work (although it remains quite vague/confusing, see next).

Great care has been taken in a correct mathematical formulation off all definitions and proofs.

The authors have verified some of their theoretical claims with a series of experiments.

**Weaknesses:**

1.	From your abstract, introduction and conclusion, it is not clear why you would study  co-correlation between layers in the first place, and/or how this relates to robustness and generalization. E.g., starting from line 40, the reader can infer: Dirichlet energy = measure of adversarial robustness -> Dirichlet energy per layer -> co-correlation -> gradient descent increases co-correlation. But how and why is this co-correlation exactly related to adversarial robustness? Etc.

2.	This does not become really clear to me in the other parts of the paper either. I understand that the dirichlet can be used to measure the gap between generalization and adversial risk. But what this means for individual layers and why you would want to look into this remains unclear. E.g., Line 163: “Since the Dirichlet energy defined in Equation (7) can be used to measure the variability of mappings,it follows that this metric could be employed to evaluate the adversarial robustness of individual layers or modules within neural networks. Consequently, this allows for an assessment of whether there is collaboration between these components in terms of adversarial robustness.”: this seems an important point, but I could’t understand what you meant exactly, especially the second sentence.

3.	L117: why is the second layer random and fixed? If needed for a mathematical analysis, what is the influence of this on the generality of your findings?

4.	The claim made in L160-161 cannot be supported by fig. 1 alone. It’s not because Dirichlet energy has a similar (inverse) profile over width and initialization as the adversarial robustness, that one influences the other directly. The same goes for the other figures.  More information/verification seems needed.

5.	L192-194: what about non-linear activation functions?

6.	The experiments are only conducted on the simplified models used for the theoretical derivations, and as such do not show anything about how the theoretical findings relate to actual, practical neural networks (i.e., models with more layers, where no layers are held fixed).

Typos/grammar/symbols:
7.	Line 64, 76, 77, 105, 111 (strange sentence), 127, caption figure 1, 275, …
8.	Eq 1: f_W undefined
9.	Fig 1: legend and labels too small, caption not clear

**Questions:**

See points raised above. In general, I think this work could be a valuable contribution, but the quality of the text has to be increased to better motivate and explain the approach. In it’s current state, the work is hard to judge.

If you think a change to the manuscript is warranted, please specify how you would update it.

**Limitations:**

The limitations are not addressed sufficiently. E.g., not clear what the influence of the strong assumption that the second layer remains fixed has on the generality of the findings.

No potential negative societal impact.

---

> ### Author Rebuttal · Authors · 2024-08-07
>
> Thank you very much for your valuable feedback and careful review. Your detailed examination of the paper and identification of its flaws are greatly appreciated. My rebuttals are as follows:
> ## The reason why study co-correlation:
> The motivation for studying co-correlation actually comes from the research question:
>
> *Can different layers of a neural network collaborate against adversarial examples?*
>
> This is an important point, especially in an era dominated by complex neural networks. For example, in the main block of a vanilla Vision Transformer, we have Multi-head Self Attention (MSA) followed by an MLP. People may expect these components to have specific functionalities and imagine the forward propagation working like independent robots on a conveyor belt. However, even for a simple MLP, there is hardly any research on whether different layers interact with each other, let alone on the effects on adversarial robustness. Our work aims to fill this gap to some extent.
> ## How co-correlation related to adversarial robustness:
> The relationship between co-correlation and adversarial robustness is bridged by the Dirichlet energy, as depicted in Theorem 4.5 and Eq. (15). While co-correlation $\varrho_{\boldsymbol{\phi}, \boldsymbol{\varphi}}$ is not directly related to adversarial robustness, it is the most significant factor affecting the Dirichlet energy as shown in Thm. 4.1. Specifically, let $\phi \circ \varphi$ be a 2-layer neural network where $\varphi$ represents the first layer and $\phi$ represents the second layer. Since the terms $\Big(1 + \frac{var_{\boldsymbol{\phi}, \boldsymbol{\varphi}}}{\mu^2_{\boldsymbol{\phi}, \boldsymbol{\varphi}}}\Big)^{\frac{1}{2}}$ is negligible, the co-correlation $\varrho$ dominate how the Dirichlet energy of each layer can be passed to that of neural network.
> ## About L163:
> The use of Dirichlet energy to measure adversarial robustness was first considered in the paper by Dohmatob et al. [8], as mentioned in line 137. It was originally proposed to replace the Lipschitz constant as a better measure of adversarial robustness, as shown on page 14, Appendix A, in [8]. In our work, we first demonstrate that Dirichlet energy can measure the gap between generalization and adversarial risk, implying that Dirichlet energy is a proper measurement for adversarial robustness since higher Dirichlet energy indicates a larger gap.
>
> With simple modifications, we can also calculate the Dirichlet energy for a single layer or blocks within neural networks, as defined in Eq. (7) in Def. 3.2. We use this concept because of the decomposition of the Dirichlet energy for neural networks, as shown in Eq. (15). If we regard the neural networks as compound functions, i.e., $\theta \circ \phi$, the decomposition of the Dirichlet energy for this compound function requires the computation of Dirichlet energy for $\phi$ and $\theta$, each of which can be an individual layer inside the neural network. This explains the first sentence that we can evaluate the adversarial robustness of individual layers. I apologize for the misuse of “adversarial robustness,” which will be corrected in a later version.
>
> For the second sentence, as is shown in Eq. (15), since the term $\Big(1 + \frac{var_{\boldsymbol{\phi}, \boldsymbol{\varphi}}}{\mu^2_{\boldsymbol{\phi}, \boldsymbol{\varphi}}}\Big)^{\frac{1}{2}}$ is negligible shown in Fig. 4, leave co-correlation $\varrho$ dominating whether Dirichlet energy for each mapping can be transferred to overall Dirichlet energy. Hence, we say that we can assess the collaboration between layers via $\varrho$.
> ## About L117
> The random and fixed second layer is used for simplification of our analysis, following the previous work by Du et al. [9], which was published in ICLR. Since we use similar approaches—directly analyzing the gradient descent—and to make the results easier to understand, we adopted the same setting. We will emphasize this in the paper of a later version.
> ## About the claim in L160
> Each neural network in Fig. 1 (a) and (b) was trained to converge under identical conditions, except for width and weight initialization. Therefore they demonstrate the relationship between Dirichlet energy and adversarial robustness, though not causally. However, as verification of Thm 4.1, it is adequate to show the correctness of the proof. And more experiments will be included in a later version.
> ## About L192-L194
> L192-L194 discusses the interpretation of co-correlation in a linear case without activation functions to simplify understanding. Since the activation functions will make the transformation at layers become non-linear, we can replace the weight matrix with the Jacobian. Let $\theta$ and $\phi$ be the weight matrices with activation functions, then Eq. (11) becomes
> $$
> \frac{\Vert J_{\theta} J_{\phi}  \Vert_2 }{\Vert J_{\theta} \Vert_2 \Vert J_{\theta} \Vert_2 }
> $$
> The explanation at L190-L194 is still applicable for the Jacobians of both layers. The difference is that the Jacobian varies with different inputs, potentially attect co-correlation, as empirically shown for ReLU in Fig. 1 (c) and (d).
> ## Extra experiments
> We conduct the experiments with ResNet50 and Wide-ResNet50 on CIFAR10. We divided the ResNets in two ways and used the Adam optimizer with a learning rate of 0.003 to track the dynamics of co-correlations. The experiment results are available at the link: [Dynamic of co-correlation of the divide as $A_1$, $A_2$](https://i.imgur.com/icGOSOK.png) and [Dynamic of co-correlation of the divide as $B_1$, $B_2$](https://i.imgur.com/EfTfHx4.png). As shown in both figures, the co-correlation increases over the training epochs, although the upward trend for the separation of $A_1$ and $A_2$ is not strictly monotonic. The figure shown in the anonymous link: [Illustration of the divide for ResNet50](https://i.imgur.com/1OPWAuO.png) shows the way we divide ResNet50.
>
> ### We will check all lines and make them more clear in a later version.

---

> > ### Comment · Reviewer_DyJc · 2024-08-13
> >
> > I have carefully read the rebuttal, and I would like to thank the authors for their effort and clarifications. The additional experiments are performed with more complex models and strengthen the work. I appreciated the clarifications about the goal of the paper and the relationship between concepts.
> >
> > It's still rather hard to judge how the final manuscript will be improved, as the statements about this are rather unspecific ("We will emphasize this in the paper of a later version", " more experiments will be included in a later version "). I think the *presentation* of this paper could be greatly improved -but this might be achieved in the camera-ready version. E.g., the research goal as stated/clarified in the rebuttal is quite different from what a reader can infer from, e.g., the original abstract. But I'd like to give the authors the benefit of the doubt, and I'll raise my score to 5.

---

> > > ### Author Response · Authors · 2024-08-13
> > > **Proposed Updates to the Manuscript**
> > >
> > > Thank you very much for your response. I have carefully checked my manuscript according to your response, and I would like to propose the following updates to the manuscript:
> > > ## Abstract
> > > The main issue with the abstract is the lack of a clear statement of the primary research goal, particularly in L8 to L13. The sentences in this section will be rephrased to clearly emphasize the research goal. This part will be modified as follows: we will first present the research goal, followed by an introduction to the core concept we proposed to evaluate collaboration between layers, along with an intuitive explanation of this concept. A brief discussion of the different behaviours of over and under-parameterized neural networks will then be included, followed by our experimental results.
> > >
> > > ## Introduction
> > > The issue in the Introduction is similar to that in the abstract — the absence of a clearly stated research goal: to determine whether there is collaboration between layers in resisting adversarial examples. Additionally, it lacks a clear explanation of the relationships between different concepts, such as the connection between Dirichlet energy, adversarial robustness, and co-correlation. To address these issues, we will follow the same logic as the modification in the abstract and rephrase the third paragraph (L33 to L39), with the research goal emphasized to clearly outline the motivation for this work. An additional paragraph will be added to explain the concepts used in the paper and their relations. A clear flow of how we solve the research goal will also be included in the same paragraph.
> > >
> > > ## Preliminary
> > > A clear explanation of the neural network setting will be added. In addition to citing the setting from [9], we will explain why this setting is reasonable and whether the theoretical results can be generalized to broader neural networks. The initialization setting mentioned in line 117 will also be explained, including a discussion on the impact of deviating from this setting.
> > >
> > > ## Measure Adversarial Risk by Dirichlet Energy
> > > Additional empirical evidence using more complex neural networks will be included to verify our proposed Theorem 4.1. Experiments involving more complex models, such as ResNet50 and WRN50, will be presented in a new figure. Additionally, a brief discussion of the non-linear case will be added to the "Interpretation of Co-correlation" paragraph.
> > >
> > > ## On dynamics of co-correlation
> > > In L214 to L219 and L254, we will clarify the reasoning behind our assumptions and compare them with existing works, such as those by Lyu and Li [22], Ji and Telgarsky [18], Kunin et al. [20], and Frei et al. [13], to highlight the advantages of our simplified assumptions.
> > >
> > > ## Experiments
> > > Some of the additional experiments will be included in this section, along with brief descriptions.
> > >
> > > ## Conclusion
> > > In this section, we will rephrase the paragraph to re-emphasize our research goal, followed by an explanation of how we modelled the problem using Dirichlet energy and the proposed concept of co-correlation, supported by our experimental results. The re-emphasis of the research goal will be placed at the beginning of this section.
> > >
> > > ## Overall modification
> > > The use of the terms "under and over-parameterized neural networks" will be clarified as "narrow and wide neural networks" when referring specifically to 2-layer networks. And they will be used only when discussing more general or deeper networks. Additionally, a brief discussion on neural networks with varying depth and width but the same number of parameters will be included in the appendix, along with more experiments, and will be mentioned in the main content.
> > >
> > > Thank you again for your thorough review of the paper. Your feedback has been invaluable in helping me improve the manuscript.

---

> ### Author Response · Authors · 2024-08-12
> **A kind reminder for your response**
>
> Due to the limited time available to respond, and since I have not received any feedback on whether I answered your question, I kindly request that you let me know if my response was satisfactory. This would help me understand where I can improve my work. Thank you!

---

### Official Review · Reviewer_yLC9 · 2024-07-12

**Soundness:** 3
**Presentation:** 2
**Contribution:** 2
**Rating:** 5
**Confidence:** 3

**Summary:**

This paper investigates the implicit bias of gradient descent in over-parameterized neural networks, particularly focusing on the collaboration between consecutive layers. The study first introduces Dirichlet energy to evaluate the adversarial risk. Then it decomposes Dirichelt energy between layers and measures the alignment of feature selections between layers. The collaboration of Dirichlet energy between layers is called co-correlation. Through theoretical analysis and extensive experiments, the authors demonstrate that over-parameterized networks exhibit stronger resistance to increased co-correlation, thereby enhancing adversarial robustness.

**Strengths:**

The paper introduces the novel concept of co-correlation to quantify collaboration between layers, offering a fresh perspective on implicit bias in neural networks. The notion of co-correlation helps with distinguishing the dynamics between over-parameterized and under-parameterized networks. Moreover, this paper provides robust theoretical analysis for the properties of the dynamics of co-correlation.

**Weaknesses:**

1. The theory part of the paper mainly analyzes the dynamics of co-correlation. However, the paper only explain the relation between Dirichlet energy and adversarial robustness. It does not directly connect adversarial robustness with co-correlation in analysis.Without a valid analysis, co-correlation dynamics cannot lead to the conclusions on adversarial robustness.
2. In the analysis of co-correlation,  this paper proved $C(t)\>0$ after training for some time. This result indicates the increeasing trend of co-correlation but not explaining the value and the rate of the convergence, which shows the strength of the implicit bias. Moreover, it could also be better if the authors discuss in the theory part about the effect of initialization shown in experiment.

**Questions:**

1. Why the final co-correlation of MLP is smaller than linear net with same width and initialization?
2. How will the Dirichlet energy for each mapping $\phi$ and $\psi$ changes? Will this also affect the Dirichlet energy and adversarial robustness of the neural network?

**Limitations:**

The authors have addressed the limitations of their work, acknowledging the challenges in extending the approach to more complex models and the need for broader evaluations.

---

> ### Author Rebuttal · Authors · 2024-08-07
>
> Thank you for your valuable feedback and careful review. The rebuttals are as follows:
>
> ## About the relation between co-correlation and adversarial robustness
> The relationship between co-correlation and adversarial robustness is bridged by the Dirichlet energy, as depicted in Theorem 4.5 and Eq. (15), shown as
> $$
>   \mathfrak{S}(\boldsymbol{\phi}\circ \boldsymbol{\varphi}) = \varrho_{\boldsymbol{\phi}, \boldsymbol{\varphi}} \Big(1 + \frac{var_{\boldsymbol{\phi}, \boldsymbol{\varphi}}}{\mu^2_{\boldsymbol{\phi}, \boldsymbol{\varphi}}}\Big)^{\frac{1}{2}} \rho_{\boldsymbol{\phi}, \boldsymbol{\varphi}}  \mathfrak{S}(\boldsymbol{\phi}) \mathfrak{S}(\boldsymbol{\varphi}).
> $$
> While co-correlation $\varrho_{\boldsymbol{\phi}, \boldsymbol{\varphi}}$ is not directly related to adversarial robustness, it is the most significant factor affecting the Dirichlet energy and the Dirichlet energy is directly related to adversarial robustness as shown in Thm. 4.1. Specifically, let $\phi \circ \varphi$ be a 2-layer neural network where $\varphi$ represents the first layer and $\phi$ represents the second layer. First, as demonstrated in Eq. (15), the Dirichlet energy of the network $\mathfrak{S}(\phi \circ \varphi)$ is dominated by the co-correlation $\varrho$, since the terms $\Big(1 + \frac{var_{\boldsymbol{\phi}, \boldsymbol{\varphi}}}{\mu^2_{\boldsymbol{\phi}, \boldsymbol{\varphi}}}\Big)^{\frac{1}{2}}$ are negligible as shown in Fig. 4. Seondly, our work concerns more about how the Dirichlet energy of layer 1 and layer 2, i.e., $\mathfrak{S}(\boldsymbol{\phi})$ and $\mathfrak{S}(\boldsymbol{\varphi})$, can be passed to overall  Dirichlet energy of neural networks. And this interaction between layers can be quantified by co-correlation $\varrho$.
>
> It is important to note that this paper primarily aims to understand the interplay between layers concerning adversarial robustness, i.e. $\varrho$, rather than adversarial robustness itself. Therefore, the direct relationship between co-correlation and adversarial robustness is not our primary focus.
>
> ## About the value and the rate of convergence
> As indicated in Thm. 5.3 and Thm. 5.5, Eq. (17) and Eq. (20), the exact value of $C(t)$ depends on several assumptions about the inputs and the state of the weight matrix $W(t)$. It is possible to provide a more precise evaluation of $C(t)$ based on specific assumptions about the inputs. However, doing so requires particular assumptions, such as Gaussian distribution of the inputs and weight matrix.
>
> Because our work primarily aims to answer the research question of whether there exists collaboration between layers to improve adversarial robustness, we focused on qualitative analysis of dynamics change rather than quantitative analysis of the convergence speed. This is an interesting problem, but it is impossible to address all issues within a limited article. Future studies may explore this idea.
>
> ## About co-correlation of MLP is smaller than linear net
> As is shown on page 5, the co-correlation represents the feature alignment of layers in neural networks. Introducing an activation function, at least ReLU, as an 'isolation' layer between $W_1$ and $W_2$, is suggested to reduce co-correlation. This concept is experimentally verified in Fig. 1 (c) and (d). While we did not formally prove this idea, and different activation functions other than ReLU may even enhance co-correlation, our conclusion holds as long as the derivative of the activation function is bounded, as detailed in Assumption 5.4.
>
> ## The effect of Dirichlet energy for each mapping
> Yes, as shown in Eq. (15), the overall Dirichlet energy is also affected by that of individual mappings. However, since our primary focus is on the interplay between layers, this aspect is beyond the scope of our paper. Even if the Dirichlet energy for each mapping, $\mathfrak{S}(\boldsymbol{\phi})$ and $\mathfrak{S}(\boldsymbol{\varphi})$, are large, co-correlation still represents the interplay between $\phi$ and $\varphi$. Additionally, since other statistics are negligible shown in Fig. 4, whether the high value of Dirichlet energy for each mapping can be transferred to $\mathfrak{S}(\boldsymbol{\phi} \circ \boldsymbol{\varphi})$ still highly depends on $\varrho$.

---

> > ### Comment · Reviewer_yLC9 · 2024-08-13
> >
> > I thank the authors for their detailed rebuttal. I think I have to explain my concerns again since the authors thought the questions are vague and beyond the scope of their work.
> >
> > When I read this paper, I assume that the research goal of this paper is $\textbf{whether there exists collaboration between layers to improve adversarial robustness}$, which have also been mentioned by the authors in their rebuttal. In order to completely solve the problem, I think the paper should answer the following two questions:
> > 1. the existence of collaboration between layers.
> > 2. the collaboration improves adversarial robustness.
> >
> > My main concern is the completeness of this work since I was not satisfied with the answers of the two questions above. In my review, I provided two major weakness. The second weakness of $C(t)$ is about question No.1 above, where I think $C(t)>0$ is not a very strong evidence of the existence of collaboration between layers. Only proving $C(t)>0$ shows that $\varrho$ increases during training but does not rule out the case that $\varrho(t)$ only increases a little, which is not enough to claim the existence of the collaboration.
> >
> > My first major concern is directly connected with question No.2 above, where I think the paper only provides vague explanations to this question. However, in the response from the authors, they claims that this question is not their main focus. According to the general rebuttal and the responses under other reviews, I realized that the primary focus of this paper is question No.1 only. I think it would be better for the authors to address their resaerch goal more explicitly in their paper and responses.
> >
> > After reading the responses, I have no concern on the completeness since question No.2 is not a primary goal of this paper and the problem of $C(t)$ is not that significant because the authors provided some experiment evidence on question No.1 at least. However, the contribution of the paper is not that signinificant with only showing the existance of the collaboration between layers. I have to admit that this implicit bias does not rely on strong assumptions and can be extended to deep networks. But I am wondering if this implicit bias provides significant insight on understanding the benifit of GD in training neural networks without focusing on question No.2 above.
> >
> > According to the explanations above, I will keep the score. Thanks the authors again for their rebuttal and the additional results they provided.

---

> > > ### Author Response · Authors · 2024-08-13
> > >
> > > Thank you very much for your responses. Below is my response to the questions raised regarding the paper.
> > > ## About the existence of collaboration between layers
> > > The paper is structured as follows: First, in Theorem 4.1 and Figures 1(a) and 1(b), we establish a connection between adversarial risk and Dirichlet energy, showing that Dirichlet energy, denoted as $\mathfrak{S}(f)$, can approximate the gap between adversarial risk and natural risk. To facilitate understanding of Theorem 4.5, we first define our core concept, co-correlation $\varrho$, along with other related statistics. In Theorem 4.5, the Dirichlet energy of a compounded mapping $\phi\circ\varphi$ can be decomposed into the product of the Dirichlet energy for each mapping $\mathfrak{S}(\phi)$, $\mathfrak{S}(\varphi)$, co-correlation $\varrho$ and other statistics. Figure 4 in the appendix shows that these other statistics are negligible, so given that $\mathfrak{S}(\phi)$ and $\mathfrak{S}(\varphi)$ are fixed, co-correlation $\varrho$ represents the interaction between $\mathfrak{S}(\phi)$ and $\mathfrak{S}(\varphi)$ w.r.t. adversarial robustness. Here, $\phi\circ\varphi$ can be considered as a neural network, with $\phi$ and $\varphi$ as the functional components that constitute the network.
> > >
> > > Based on this flow, we define the collaboration between layers—or more broadly, between different components in neural networks—by co-correlation $\varphi$. It generally describes the interaction between $\phi$ and $\phi$ concerning adversarial robustness, where a lower value of $\varrho$ implies stronger collaboration between layers. Therefore, as long as $\varrho$ increases during GD, we can assert that the implicit bias of GD hinders collaboration between layers.
> > >
> > > In cases where $\varrho$ increases only a little, and the overall $\mathfrak{S}(\phi\circ\varphi)$ decreases due to lower values of $\mathfrak{S}(\phi)$ and $\mathfrak{S}(\varphi)$, it is still fair to claim the existence of collaboration.
> > >
> > > ## Beyond the existence of collaboration
> > > In addition to proving the existence of collaboration between layers, we also explore how the over- and under-parameterization of neural networks—i.e., wider and shallower 2-layer networks—affects adversarial robustness. As shown in Figure 3 and discussed in Section 6.2, GD tends to improve the performance of narrower neural networks by fostering co-correlation among layers, which will potentially weaken the adversarial robustness. On the other hand, over-parameterized networks (the wider ones) are trained with less reliance on interlayer correlation (a smaller $\varrho$ implies stronger collaboration), leading to inherently more robust models, while under the assumption that $\varrho$ is a more dominated factor than $\mathfrak{S}(\phi)$ and $\mathfrak{S}(\varphi)$. The measure of $\mathfrak{S}(\phi)$ and $\mathfrak{S}(\varphi)$ will be included in a camera-ready version of the paper. This observation complements the argument made by Frei et al. [13], which states that GD leads neural networks to non-robust solutions. We show that this problem can be mitigated to some extent in over-parameterized networks. GD will lead neural networks to a non-robust solution, but not the worst one for over-parameterized neural networks.
> > >
> > > Finally, the discussion on the value of $C(t)$ is indeed intriguing, particularly in the context of neural networks with different architectures. However, due to space limitations, it wasn't possible to include everything in the 9-page paper. This topic could be a promising direction for future research, where more comprehensive work will be done to explore the rate of convergence and the value of $C(t)$.

---

> > > > ### Comment · Reviewer_yLC9 · 2024-08-14
> > > >
> > > > Thanks for detailed extra explanation. I am satisfactory about the explanation and believe that this work is complete with some contribution in adversarial robustness. I will raise the score by 1.

---

> ### Author Response · Authors · 2024-08-12
> **A kind reminder for your response**
>
> Given the limited time available to respond and the lack of feedback on whether I adequately addressed your question, I kindly request that you let me know if my response was satisfactory. This would help me identify areas where I can improve my work. Thank you!
>
> Additionally, I noticed that your question is somewhat vague, and many aspects fall beyond the scope of our research. If possible, providing more specific questions would be greatly appreciated.

---

### Official Review · Reviewer_G4as · 2024-07-13

**Soundness:** 3
**Presentation:** 3
**Contribution:** 4
**Rating:** 6
**Confidence:** 4

**Summary:**

This work focuses on studying the implicit bias of correlation between intermittent layers of neural nets and uses this metric to analyze the adversarial robustness of networks in under and over parameterized regimes. The authors further use these findings to suggest that in the under parametrized case, gradient descent enhances accuracy by forcing co-correlation between layers which in turns leads to worse adversarial robustness. However, in the overparameterized case, models depend less on the interlayer correlations and thus result in more robust models.

The authors approach the problem by using the Dirichlet energy to bound the adversarial risk. They then introduce the co-correlation metric, designed to describe the alignment of layers in a linear 2-layer neural net. Further, the authors manage to bound the Dirichlet energy using the co-correlation metric and additional terms but provide experimental evidence to indicate that main term changing throughout training in under over/under parameterized scenarios is the co-correlation metric.

The paper then focuses on studying the dynamics of co-correlation change during training on the linear model with emphasis on layer width and initialization, describing how co-correlation increase happens mainly during the early stages of training. Finally, the authors provide experimental results and extend their work to general MLPs.

**Strengths:**

Overall I found the work to be quite interesting, well written and described. The ideas were well explained, assumptions are mostly reasonable and the paper is easy to understand for me. Additionally I like seeing works done in the interlayer interactions of deepnets and believe such observations and results could have further experimental and theoretical implications.

I am also quite interested in the results regarding implication of initialization and am somewhat surprised by the experimental results describing the importance of initialization.

**Weaknesses:**

1) While I understand that the focus of the work is on theory, I would like to see 1-2 more experimental results, potentially with CIFAR-10 or larger networks (maybe not MLP) just to confirm the results and give a better perspective.

2) The authors often make claims regarding over-under parametrization vs width of layers. I find it a bit confusing whether some of the claims are considering the number of network parameters or the actual width of layers and would like to see some more distinction. For example in experimental cases, it's possible to try out narrower but deeper networks with the same number of parameters as wider ones to isolate the width effect. I understand that this could be hard in such controlled cases and small MLP’s of course.

I will address some more of my concerns in the question section.

**Questions:**

1) To be clear the authors don’t provide any theoretical explanation for why the other terms used for the upper bound in Thm 4.5 are negligible ? And on that note, am I correct in assuming that the point of Fig 4 in appendix is to show how the variance/mean and linear correlation don’t change much with model width ? If so, shouldn't you consider different initializations as well ?

2) On equation 19, the authors make the claim that the first term in the lower bound is large early on ? Wouldn’t this also be true during the late stage of training since the learning rate is smaller and $ \tilde{x} $ stabilizes ? With enough training epochs, I expect that to be the case right ?

3) Could authors provide further explanation on the potential impact of weight regularization and co-correlation given Property 2 ?

4) The comment made on line 267, wouldn’t it be the opposite that $ \tilde{x}_* $ fluctuates more during the early stages of training with higher learning rate, as mentioned previously for Thm 4.5 ?

---

> ### Author Rebuttal · Authors · 2024-08-07
>
> Thank you very much for your valuable feedback and careful review, you indeed checked the details of the paper and pinpointed the flaws. It is highly valued feedback. My rebuttals are as follows:
>
> ## More experiments
> ### Experiments on ResNet50 and Wide-Resnet50
> We conducted additional experiments on larger and deeper networks, specifically ResNet50 and Wide-ResNet50 on CIFAR10. These experiments will be included in the paper in a later version. In the figures shown in the anonymous links: [Illustration of the divide for ResNet50](https://i.imgur.com/1OPWAuO.png), we divided the ResNets in two ways and used the Adam optimizer with a learning rate of 0.003 to track the dynamics of co-correlations. The experiment results are available at [Dynamic of co-correlation of the divide as $A_1$, $A_2$](https://i.imgur.com/icGOSOK.png) and [Dynamic of co-correlation of the divide as $B_1$, $B_2$](https://i.imgur.com/EfTfHx4.png). As shown in both figures, the co-correlation increases over the training epochs, although the upward trend for the separation of $A_1$ and $A_2$ is not strictly monotonic.
>
> ### About varying depth and width
> Your suggestion to conduct experiments on networks with varying depth and width, but the same number of total parameters, is quite interesting. However, in this paper, we focus on the theoretical analysis of whether there is collaboration between layers against adversarial examples. How different architectures of neural networks impact the co-correlation, especially with varying depth and width, will be considered in future work.
>
> ### Negligible of other terms on different weight initialization
> Yes, Fig. 4 in the appendix demonstrates that the other terms are negligible. Additional experiments with different initializations are available at [Relative std](https://i.imgur.com/dLOv5yv.png) and [Linear correlation](https://i.imgur.com/GzsDl3G.png). We consider the MLP with a width of 512 with varying weight initializations. Similar to Fig. 4 in the appendix, the first and second figure show the $\frac{var^{\frac{1}{2}}}{\mu}$ and $\rho$ in Eq. (15). Empirically, it shows that $\Big(1 + \frac{var_{\boldsymbol{\phi}, \boldsymbol{\varphi}}}{\mu^2_{\boldsymbol{\phi}, \boldsymbol{\varphi}}}\Big)^{\frac{1}{2}} \rho_{\boldsymbol{\phi}, \boldsymbol{\varphi}}$ is quite close to 1, making it negligible in the Eq (15).
>
> ## About Eq.(19) and training stage
> According to Eq. (19), the first term of Eq. (19) is
> $$
> \frac{\sum_{\tau=1}^{t}\widetilde{\boldsymbol{x}}(\tau)^T\widetilde{\boldsymbol{x}}(t)}{\Vert W(t) \Vert_2^2}\cdot \Big(1 - \big(\boldsymbol{v}(t)^T\boldsymbol{a}\Big)^2\Big)
> $$
> The exact value of this term is complex; however, the sign of this term depends on $\sum_{\tau=1}^{t}\widetilde{\boldsymbol{x}}(\tau)^T\widetilde{\boldsymbol{x}}(t)$, since other terms are positive, as explained in Thm 5.3. According to Eq. (17), $\widetilde{\boldsymbol{x}}$ is a weighted average of all inputs, where the weight is the difference between the ground truth $y_i$ and the predicted likelihood $sig(u_i(t))$.
>
> At the initial stage with a small learning rate, $\widetilde{\boldsymbol{x}}(\tau_1)$ is similar to $\widetilde{\boldsymbol{x}}(\tau_2)$ for $\tau_1, \tau_2 \in[t]$. More specifically, let $t$ be 5 epochs. Since the model hasn't learned much, the predicted likelihood $sig(u_i(t))$ of $\tau_1 = 1$ and $\tau_2 = 4$ are all similar to each other; therefore, $\widetilde{\boldsymbol{x}}(\tau_1)$ and $\widetilde{\boldsymbol{x}}(\tau_2)$ are similar in terms of cosine similarity.
>
> However, in the later stages, as the model learns more from the data, most of the predictions become correct, and $y_i - \sigma(u_i)$ will approach zero, causing $\sum_{\tau=1}^{t}\widetilde{\boldsymbol{x}}(\tau)^T\widetilde{\boldsymbol{x}}(t)$ to converge. Continued training will also enlarge $\Vert W(t) \Vert_2^2$ and make $\Big(1 - \big(\boldsymbol{v}(t)^T\boldsymbol{a}\big)^2\Big)$ approach zero. Considering all the impacts, the co-correlation will increase very fast at the beginning, then the rate of increase will become smaller, and in the final stage, it will approach zero, as shown in Fig. 2 (c) and (d). In Fig. 2 (c) and (d), all the co-correlation of networks flattened at the end.
>
> ## About weight regularization and co-correlation on Property 2
> As shown in Eq. (19), the co-correlation is inversely related to the $L_2$-norm of the weight matrix. This $L_2$-norm is the operator norm, which is different from the norm used in weight regularization, such as L2 regularization, which is the Frobenius norm (F-norm). This implies that merely controlling the F-norm of the weight matrix may not have a direct impact on $\Vert W \Vert_2$. However, since all norms are equivalent in finite-dimensional vector spaces, L2 regularization will somehow restrict its increase during training. Additionally, since the value of $C(t)$ also depends on $\Big(1 - \big(\boldsymbol{v}(t)^T\boldsymbol{a}\big)^2\Big)$, where $\boldsymbol{v}$ is the maximal singular vector of $W(t)$, it is difficult to determine whether $C(t)$ will become larger because of weight regularization.
>
> ## About L267
> Yes, if the learning rate is too large, whether at the initial stage or at any stage, $\widetilde{x}^{\star}$ may fluctuate too much, but it may also prevent the model from converging or result in poor performance. In such cases, it is possible that $C(t)$ could become negative. However, in most machine learning settings, it is required that the learning rate not be too large, such as using 3e-4 with Adam, to ensure convergence and good performance. Therefore, I believe the assumption that the learning rate is not too large is realistic.

---

> ### Author Response · Authors · 2024-08-12
> **A kind reminder for your response**
>
> Due to the limited time available to respond, and since I have not received any feedback on whether I answered your question, I kindly request that you let me know if my response was satisfactory. This would help me understand where I can improve my work. Thank you！

---

> > ### Comment · Reviewer_G4as · 2024-08-13
> >
> > Thank you for the detailed response and I apologise for the late reply. First I would like the appreciate the authors for including additional experimental results and responding to my concerns, specifically with question 2 and 4. I belive this work has an intriguing concept and solid theoretical work so I would like to keep my current evaluation.

---

> > > ### Author Response · Authors · 2024-08-13
> > >
> > > Thank you so much for your review and response.

---

### Official Review · Reviewer_KqkQ · 2024-07-15

**Soundness:** 3
**Presentation:** 2
**Contribution:** 2
**Rating:** 5
**Confidence:** 3

**Summary:**

In this work, the authors study the tradeoff between generalization and adversarial robustness from the perspective of collaboration between the layers of a neural network (NN). They adapt the concept of Dirichlet energy to analyze the robustness of different layers in the network. Decomposing this across layers allows them to quantify the alignment between feature selection between consecutive layers as collaboration correlation (or co-correlation). They show that the co-correlation increases while training, which leads to lower adversarial robustness, MLPs with larger widths exhibit more resistance to increased co-correlation and present supporting experimental results.

**Strengths:**

This work presents a novel *view* on the tradeoff between adversarial robustness and generalization of NNs trained with gradient descent.

**Weaknesses:**

- While the results are somewhat interesting, it is unclear what the implications of these results are, since many of the observations in this work have been noted in prior work, and are not surprising. It is also not clear if the proposed concept of co-correlation can be used can be used for other applications .

- There are several typos/grammatical errors/formatting issues that should be corrected:
    - In line 56, ‘adversarial’ should be ‘adversarially’.
    - The last part of lines 64-65 should be rephrased.
    - In line 77, ‘digger’ should be ‘dig’.
    - The heading of section 3 should be ‘Preliminaries’.
    - Some of the notations (e.g., $sig$) should be corrected (to sig).
    - In line 127, ‘Slimier’ should be ‘Similar’.
    - The formatting of Theorem 4.1 (lines 155-156) should be corrected.
    - In line 179, $J\mathbf{\psi}$ should be $J_{\mathbf{\psi}}$.
    - In Ass. 5.2, L-2 should be $L_2$.
    - In Prop. 1, ‘flattened’ should be ‘saturated’.
    - A reference is missing in line 275.
    - Line 297 should be rephrased.
    - Missing ‘.’ in line 311.

**Questions:**

Please see the weaknesses section.

**Limitations:**

There are no societal negative impacts of this work.

---

> ### Author Rebuttal · Authors · 2024-08-07
>
> Thank you for your valuable feedback and careful review. We appreciate the opportunity to clarify and expand on our work. Our rebuttals are as follows:
>
> ## Theoretical Implications
> Although it may not be immediately apparent in the paper, the implications of our work are significant. From a theoretical perspective on implicit bias research, our approach is distinct in assuming only L-2 norm-bounded inputs, Gaussian weight initialization, and bounded derivatives of the activation functions. This differs from most works on implicit bias, such as those by Lyu et al. [23], Frei et al. [12], Kunin et al. [20], and Frei et al. [13], where proofs are typically provided for 2-layer neural networks and rely on strong assumptions on inputs, making it difficult to generalize to deep neural networks. The extension to deeper neural networks in these works may require the same strong assumptions on features between layers, which is unrealistic. However, due to our assumptions on inputs needing only to be $L_2$-norm-bounded, our work can be more easily extended to deep networks. This extension was not included in the paper due to page limitations. However, we have provided experiments with ResNet50 and Wide-ResNet50 on CIFAR10 in this rebuttal and the later version of this paper.
>
> In the figures shown in the anonymous links: [Illustration of the divide for ResNet50](https://i.imgur.com/1OPWAuO.png), we divided the ResNets in two ways and used the Adam optimizer with a learning rate of 0.003 to track the dynamics of co-correlations. The experiment results are available at [Dynamic of co-correlation of the divide as $A_1$, $A_2$](https://i.imgur.com/icGOSOK.png) and [Dynamic of co-correlation of the divide as $B_1$, $B_2$](https://i.imgur.com/EfTfHx4.png). As is shown in both figures, the co-correlation increases over the training epochs, although the upward trend for the separation of $A_1$ and $A_2$ is not strictly monotonic.
>
> Additionally, our work offers a possible explanation for the observation that wider neural networks are more resilient to adversarial attacks. This complements recent findings by Frei et al. [13], who suggest that implicit bias leads to non-robust solutions. Our work suggests that these non-robust solutions can be mitigated by increasing the network width.
>
> ## Practical Implications
> Practically, our findings imply that designing an isolation layer to decouple the correlation between layers could enhance adversarial robustness. However, due to page limitations, we concentrate on the theoretic analysis of whether there is a collaboration between layers against adversarial examples. This aspect of how to reduce the co-correlation will be explored in future works.
>
> We greatly appreciate your careful review. All typographical and grammatical errors will be corrected in a subsequent version. And all the suggestions will be considered. Thank you for your understanding and feedback.

---

> ### Author Response · Authors · 2024-08-12
> **A kind reminder for your response**
>
> Due to the limited time available to respond, and since I have not received any feedback on whether I answered your question, I kindly request that you let me know if my response was satisfactory. This would help me understand where I can improve my work. Thank you!

---

> > ### Comment · Reviewer_KqkQ · 2024-08-13
> >
> > Thank you for the detailed response. It would be helpful to include more discussion on the implications in the paper. Reading through other reviews and responses, it seems that making changes such as adding more experiments as well as intuitions about the theorems are changes that go beyond minor updates. However, the paper presents a promising viewpoint. Hence, I will maintain my score.

---

> > > ### Author Response · Authors · 2024-08-13
> > >
> > > Thank you very much for your response.

---

### Author Rebuttal · Authors · 2024-08-07

We sincerely thank all the reviewers for their careful review.

## Motivation of the paper
This paper is mainly to address the research problem:

*Whether there a collaboration between layers against adversarial examples during Gradient Descent?*

To quantify this collaboration, we introduce a new concept called *Co-correlation*, interpreted as the alignment of feature selections to maximize outputs for each layer and investigate the implicit bias of gradient descent. We theoretically show that Gradient Descent enhances this collaboration between layers. Additionally, we observe different behaviours for under- and over-parameterized neural networks: under-parameterized networks tend to foster co-correlation among layers to improve performance, whereas over-parameterized networks' performance improvement does not heavily rely on establishing such co-correlation.

## How we will adjust in the later version
After reviewing all the feedback, we will adopt all the necessary suggestions. To facilitate understanding of our work, we will add more intuition behind our theorems and better clarify our arguments. Because this is theoretical work, it inevitably has some limitations in the experimental aspects. However, more real-world experiments will be included in a later version. The code for these experiments will also be made publicly available.

---

### Decision · Program_Chairs · 2024-09-25

**Decision:**

Accept (poster)

**Comment:**

This paper explores the tradeoff between generalization and adversarial robustness, particularly focusing on the collaboration between layers. Specifically, the concept of "Co-correlation" is introduced to quantify the alignment of feature selections across layers. By decomposing the Dirichlet energy and analyzing interactions between layers, the authors reveal that gradient descent optimization enhances collaboration correlation, affecting adversarial robustness. Through theoretical analyses and experiments, this paper demonstrates how larger widths in MLPs can enhance adversarial robustness by resisting increased co-correlation.

This paper received divergent scores before the rebuttal and the reviewers raised concerns about insufficient experimental results, unclear claims, weak presentation, and limited generality. Reviewers and AC appreciate the hard work authors have put into responses during the rebuttal, including additional results and detailed clarification. The discussion led to several score increases and all the reviewers came to a consensus of acceptance. AC concurs with the evaluations given by the reviewers.

Note that half of the reviewers mentioned that the presentation needs improvement. The authors are strongly encouraged to improve the final version according to the feedback from all the reviewers as promised in the discussion.